# A community-science approach identifies genetic variants associated with three color morphs in ball pythons (*Python regius*)

**Autumn R. Brown, Kaylee Comai, Dominic Mannino, Haily McCullough, Yamini Donekal, Hunter C. Meyers, Chiron W. Graves\*, Hannah S. Seidel◉\*, The BIO306W Consortium¶**

Department of Biology, Eastern Michigan University, Ypsilanti, MI, United States of America

¶ The BIO306W Consortium comprises students in an undergraduate laboratory course at Eastern Michigan University. Students are listed in the Acknowledgments
\* cgraves6@emich.edu (CWG); hseidel@emich.edu (HSS)

**Data Availability Statement:** All relevant data are within the paper and its Supporting Information files. DNA sequences are available from GenBank, under accession numbers MZ269492-MZ269502.

## Abstract

Color morphs in ball pythons (*Python regius*) provide a unique and largely untapped resource for understanding the genetics of coloration in reptiles. Here we use a community-science approach to investigate the genetics of three color morphs affecting production of the pigment melanin. These morphs—Albino, Lavender Albino, and Ultramel—show a loss of melanin in the skin and eyes, ranging from severe (Albino) to moderate (Lavender Albino) to mild (Ultramel). To identify genetic variants causing each morph, we recruited shed skins of pet ball pythons via social media, extracted DNA from the skins, and searched for putative loss-of-function variants in homologs of genes controlling melanin production in other vertebrates. We report that the Albino morph is associated with missense and non-coding variants in the gene *TYR*. The Lavender Albino morph is associated with a deletion in the gene *OCA2*. The Ultramel morph is associated with a missense variant and a putative deletion in the gene *TYRP1*. Our study is one of the first to identify genetic variants associated with color morphs in ball pythons and shows that pet samples recruited from the community can provide a resource for genetic studies in this species.

## Introduction

Color patterns are distinctive and beautiful features of many animal species. Among the many functions of color are to camouflage animals to their surroundings, protect tissues from ultraviolet radiation, warn predators of poisons, and provide signals for mating and social communication [1]. Color patterns have also been targeted by artificial selection to create novel and fanciful color patterns in domestic animals [2].

Colors are produced through a combination of chemical pigments and physical structures (structural colors). Pigments absorb light, whereas structural colors are created when light is reflected by the nanoscale geometry of a tissue. Common pigments include brown-to-black melanin, present throughout the animal kingdom, and yellow-to-red carotenoids and pteridines, common in birds, reptiles, and lower vertebrates [3]. Color-producing structures

**Funding:** This work was supported by a Faculty Research Fellowship and James H. Brickley Award from Eastern Michigan University to HSS and an Undergraduate Research Stimulus Program Award and a Don Brown and Meta Hellwig Undergraduate Research Award from Eastern Michigan University to ARB. The funders had no role in study design, data collection and analysis, decision to publish, or preparation of the manuscript.

**Competing interests:** The authors have declared that no competing interests exist.

include ordered keratin matrices in bird feathers [4], chitin layers in butterfly wings [5], and purine crystals in the skin of fish, amphibians, and reptiles [6,7]. These color-producing structures can produce a variety of colors, including iridescent colors, depending on the wavelengths of light they reflect [8].

The genetics and development of color patterns in vertebrates have been studied most extensively in mammals. Mammals largely rely on a single type of pigment (melanin), produced in the skin by a single type of cell (melanocyte) [9]. This system provides mammals with skin and hair colors ranging from black to reddish-brown, depending on the chemical structure of the melanin [10].

Skin colors outside mammals are more diverse and include bright colors of all hues. This increased complexity is produced through a combination of structural colors, melanin pigments, and non-melanin pigments, and it relies on multiple types of specialized color-producing cells in the skin [11–15]. Color cell development has been characterized to some extent in fish [16–18], and genes controlling the use of non-melanin pigments have been identified in a few species of fish and birds [19–25]. The genetics and development of non-mammalian color patterns are less well understood in other vertebrates, particularly in reptiles [although see 26–29].

A unique resource for understanding the genetics of color patterns in reptiles is the ball python (*P. regius*). Ball pythons are native to sub-Saharan Africa, but have become common as pets in the United States [30]. Wild ball pythons exhibit a mottled color pattern, consisting of brown-to-black melanin and red-to-yellow (non-melanin) pigments in the skin (Fig 1). Captive-bred ball pythons, by contrast, include many variants of the normal color pattern [26,31–33]. These variants, referred to as 'color morphs', include animals having reduced melanin (e.g. Albino), increased melanin (e.g. Cinnamon), reduced red-to-yellow coloration (e.g. Axanthic), or complex changes in the placement of color patches on the skin (e.g. Spider, Clown, Genetic Stripe, and Enchi). Many color morphs are heritable and show simple dominant or recessive patterns of inheritance. These inheritance patterns are consistent with single-gene causality, but only a single genetic variant associated with a ball python color morph (Piebald) has been identified to date [26]. Ball pythons therefore represent a tractable yet largely untapped resource for understanding the genetics of coloration in reptiles. An additional feature of ball pythons convenient for genetic studies is that DNA samples can be obtained non-invasively from shed skin [34].

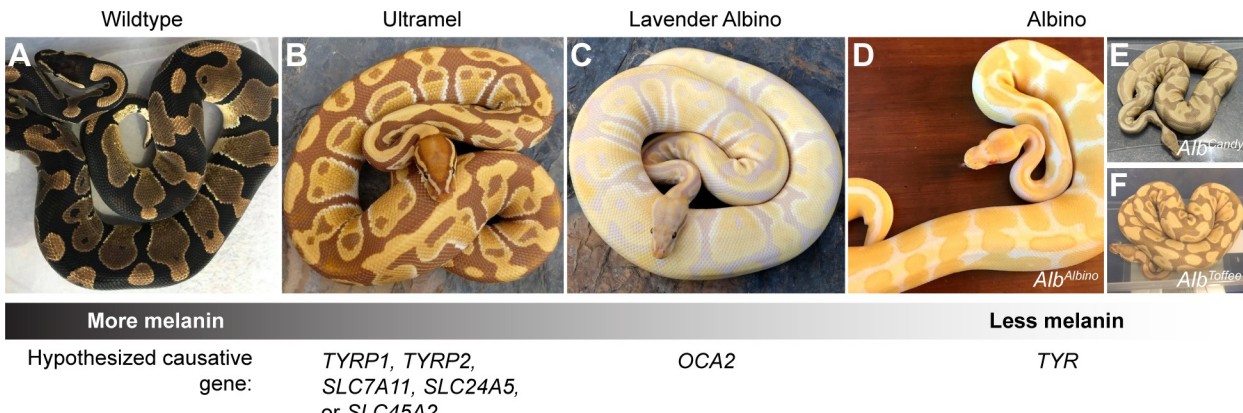

**Fig 1. The Albino, Lavender Albino, and Ultramel color morphs have reduced brown-to-black coloration, characteristic of a loss of melanin.** Red-to-yellow coloration is unaffected in these morphs. Hypothesized causative genes represent genes in which loss-of-function variants in other vertebrates produce similar phenotypes (Table 1). (A) Wildtype. (B) Ultramel. (C) Lavender Albino. (D-F) Albino. Phenotypes within the Albino color morph are variable, with some animals having skin patches that are white and others having skin patches that are light beige. (D) Albino animal described as an $Alb^{Albino}$ homozygote. (E) Albino animal described as an $Alb^{Candy}$ homozygote. (F) Albino animal described as an $Alb^{Toffee}$ homozygote. Photo credits, Ryan Young of Molecular Reptile, Chiron Graves, Phil Barclay, Michael Freedman of The Florida Reptile Ranch.

The goal of the current study was to perform proof-of-concept experiments to identify the genetic causes of color morphs in ball pythons, using shed skins of pet ball pythons recruited via social media. We focused on three morphs for which candidate genes could be readily identified: Albino, Lavender Albino, and Ultramel. These morphs show a loss of brown-to-black coloration in the skin and eyes, characteristic of a defect in melanin production (Fig 1). These morphs are recessive and non-allelic (i.e. crosses between these morphs yield offspring with normal coloration). Their loss of melanin ranges from severe (Albino) to moderate (Lavender Albino) to mild (Ultramel). This range of phenotypes mirrors the range of phenotypes observed for loss-of-function variants in genes required for melanin production in other vertebrates (Table 1). This similarity provided a list of candidate genes that we predicted might harbor loss-of-function variants causing the Albino, Lavender Albino, and Ultramel color morphs in ball pythons. Our study demonstrates the feasibility of using community-sourced samples for genetic studies in ball pythons and lays the groundwork for future investigations of reptile-specific color patterns in this species.

## Results

### Obtaining a reference sequence for melanogenesis genes in a wildtype ball python

Genes required for melanin production have been identified in humans and other vertebrates (Table 1). These genes encode enzymes that synthesize melanin (*TYR*, *TYRP1*, *TYRP2*) [87], a

**Table 1. Loss-of-function phenotypes of melanogenesis genes across vertebrates.**

| Gene | | Species with identified loss-of-function variants | Role of protein | Loss-of-function phenotype |
|---|---|---|---|---|
| *TYR* | Fish | Japanese carp [35], Japanese rice fish [36] | Rate-limiting enzyme in melanin synthesis pathway | Severe loss of melanin |
| | Frogs and snakes | Japanese wild frogs (three species) [37], Japanese rat snake [38] | | |
| | Birds | Chicken [39–41] | | |
| | Ungulates | Water buffalo [42], Cattle [43], Red deer [44], Asinara donkey [45] | | |
| | Rodents | House mouse [46], Wistar rat [47], Domestic guinea pig [48] | | |
| | Primates | Crab-eating macaque [49], Capuchin monkey [50], Hamadryas baboon [51], Human [reviewed in 52] | | |
| | Other mammals | Humpback whale [53], Ferret [54], American mink [55], Domestic cat [56–58], Silver fox [59] | | |
| *OCA2* | Fish | Mexican cavefish [60], Lake Malawi cichlid [61] | Cl⁻ channel regulating melanosome pH | Moderate loss of melanin |
| | Snakes | Corn snake [62] | | |
| | Mammals | Domestic dog [63], Bama miniature pig [64], House mouse [65], Human [reviewed in 52] | | |
| *TYRP1* | Fish | Zebrafish [66] | Enzymes contributing to melanin synthesis | Mild loss of melanin |
| | Birds | Saker falcon [67] | | |
| | Mammals | Chinese indigenous pig [68], Liangshan pig [69], Domestic dog [reviewed in 70], Domestic cat [58,71], Soay sheep [72], Valais Red sheep [73], American mink [74], Human [reviewed in 52] | | |
| *TYRP2* | Mammals | House mouse [75,76] | | |
| *SLC7A11* | Mammals | House mouse [77] | Solute transporters | |
| *SLC24A5* | Fish | Zebrafish [78] | | |
| | Mammals | Horse [79], Human [reviewed in 52] | | |
| *SLC45A2* | Fish | Japanese rice fish [80] | | |
| | Birds | Chicken and Japanese quail [81] | | |
| | Mammals | Domestic dog [82], Horse [83], House mouse [84], Bengal tiger [85], Western lowland gorilla [86], Human [reviewed in 52] | | |

**Table 2. Comparison of ball python and Burmese python genomic and protein sequences for melanogenesis genes.**

| Region | Gene | Total nucleotide sequence aligned (bp) | Nucleotide identity (%) | Amino acid identity (%) |
|---|---|---|---|---|
| Coding | TYR | 1,591 | 98.5 | 99.6 |
| | TYRP1 | 1,584 | 99.2 | 99.6 |
| | TYRP2 | 1,473 | 98.3 | 99.1 |
| | OCA2 | 2,577 | 98.9 | 99.7 |
| | SLC7A11 | 1,506 | 98.7 | 99.7 |
| | SLC24A5 | 1,379 | 98.2 | 99.5 |
| | SLC45A2 | 1,557 | 97.2 | 98.9 |
| Non-coding | TYR | 577 | 96.9 | (not applicable) |
| | TYRP1 | 1,160 | 97.0 | |
| | TYRP2 | 1,060 | 97.9 | |
| | OCA2 | 2,154 | 98.5 | |
| | SLC7A11 | 3,407 | 97.9 | |
| | SLC24A5 | 1,335 | 97.9 | |
| | SLC45A2 | 999 | 97.0 | |

chloride channel required for maintaining the pH of melanosomes (*OCA2*) [88], and transporters thought to import solutes into the cell or into organelles (*SLC7A11*, *SLC24A5*, and *SLC45A2*) [77,89,90]. These genes are highly conserved among vertebrates and occur in single copy in most vertebrate genomes. Loss-of-function variants in these genes in humans cause a genetic disorder known as oculocutaneous albinism, which is characterized by loss of melanin in the skin, hair, and eyes [52]. This loss of melanin ranges from severe to mild, depending on the causative gene and genetic variants therein [52]. Similar phenotypes occur in other animals, where the loss of melanin extends to feathers, scales, and fir (Table 1).

To obtain a reference sequence for melanogenesis genes in ball python, we amplified and sequenced the coding regions of *TYR*, *TYRP1*, *TYRP2*, *OCA2*, *SLC7A11*, *SLC24A5*, and *SLC45A2* from a single ball python having normal coloration (henceforth 'wildtype'). Primers for amplification were designed against the genome of Burmese python (*Python bivittatus*), the closest relative of ball python for which genome sequence was available [91]. Comparison of ball python sequences to sequences from Burmese python revealed 97.2–99.2% nucleotide identity within coding regions and 96.9–98.5% nucleotide identity within flanking non-coding regions (Table 2). This analysis provided a reference sequence for genes in ball python that might harbor loss-of-function variants causing the Albino, Lavender Albino, and Ultramel color morphs.

## The Albino color morph is associated with three haplotypes of *TYR*

The Albino color morph in ball pythons is characterized by an absence or near absence of melanin in the skin and eyes—the brown-to-black coloration observed in wildtype is absent or severely reduced, and skin patches appear white or light beige (Fig 1D–1F). The Albino color morph is described by breeders as having three alleles ($Alb^{Albino}$, $Alb^{Candy}$, and $Alb^{Toffee}$), although prior to the current study, it remained unclear whether these alleles represented distinct molecular variants of the same gene or the same molecular variant discovered independently three times. We began our analysis of the Albino color morph by treating all animals within the color morph as a single group.

We hypothesized that the Albino color morph was caused by loss of function of *TYR*, which encodes the enzyme catalyzing the rate-limiting step of melanin production. Loss of this

enzymatic activity causes a severe loss of melanin [87], which is typically more severe than the loss of melanin caused by loss of function of other melanogenesis genes (Table 1). To test whether the Albino color morph was associated with variants in *TYR*, we performed a small-scale association study using 50 Albinos and 56 Non-Albinos, recruited from a total of 18 states in the United States. Polymorphic sites were identified through pilot sequencing of a subset of animals (see Methods). Six polymorphic sites were selected for the association study, two near the 5' end of *TYR* and four near the 3' end (Fig 2A). Each animal was genotyped at each site, and haplotypes were reconstructed using PHASE [92,93]. Association between haplotype and color morph was tested using the case-control test of PHASE. As a negative control, we also tested for association between haplotype and color morph at *TYRP1*, *TYRP2*, and *OCA2*, using two polymorphic sites per gene (S1 Table). This analysis revealed a significant association between haplotype and color morph at *TYR* (p = 0.04, Bonferroni corrected), but no association for the other three genes (p > 0.05, Bonferroni corrected) (Fig 2A). The association for *TYR* was driven by two features of the haplotype distribution: (i) haplotype diversity was reduced from a total of 15 *TYR* haplotypes among Non-Albinos to three haplotypes among Albinos (Fig 2A and 2C), and (ii) two of the three *TYR* haplotypes found in Albinos were rare among Non-Albinos (Fig 2A). These results demonstrate an association between the Albino color morph and variants in *TYR*; further, they show that all Albino animals were homozygous (34 animals) or compound heterozygous (16 animals) for any of three haplotypes of *TYR*.

We hypothesized that each of the three haplotypes of *TYR* found among Albinos might carry a distinct loss-of-function variant in the gene. To search for such variants, we selected one Albino animal homozygous for each haplotype and sequenced the *TYR* coding regions and adjacent splice sites in these animals. We found that one of these animals was homozygous for a missense variant in the third coding region of *TYR* (MZ269492:c.A3695G), which leads to an aspartic acid-to-glycine exchange [UPQ41330.1:p.(Asp394Gly)]. This variant is termed hereafter *D394G*. A second animal was homozygous for a different missense variant, also in the third coding region of *TYR* (MZ269492:c.C3665T), which leads to a proline-to-leucine exchange [UPQ41330.1:p.(Pro384Gly)]. This variant is termed hereafter *P384L*. Both variants alter conserved residues (Fig 2B), and the *P384L* variant occurs at the same site as a similar variant (*P384A*) associated with oculocutaneous albinism in humans [94]. The third animal carried no coding variants and no splice-site variants compared to wildtype. These results show that two of the three *TYR* haplotypes found among Albinos carried missense variants that are likely disruptive for TYR protein function. We term these *TYR* haplotypes $TYR^{D394G}$ and $TYR^{P384L}$. We term the third Albino haplotype, which lacked coding or splice-site variants compared to wildtype, $TYR^{Albino}$.

We hypothesized that *D394G*, *P384L*, and an unidentified variant on the $TYR^{Albino}$ haplotype were causative for the Albino color morph. This hypothesis predicted that all Albinos would be homozygous or compound heterozygous for these variants, whereas Non-Albinos would be heterozygotes and non-carriers. To test this prediction, we genotyped our full panel of 50 Albinos and 56 Non-Albinos for the missense variants *D394G* and *P384L*. Consistent with our prediction, we found that the Albinos were exclusively homozygous or compound heterozygous for *D394G*, *P384L*, or the $TYR^{Albino}$ haplotype (Fig 2C). The Non-Albinos were exclusively heterozygotes or non-carriers (Fig 2C). The most common haplotype among Albinos was $TYR^{Albino}$ (haplotype frequency of 71%), followed by $TYR^{P384L}$ (16%) and $TYR^{D394G}$ (13%). We conclude that *D394G*, *P384L*, and an unidentified variant on the $TYR^{Albino}$ haplotype are likely causative for the Albino color morph. Any combination of these variants produces the Albino phenotype.

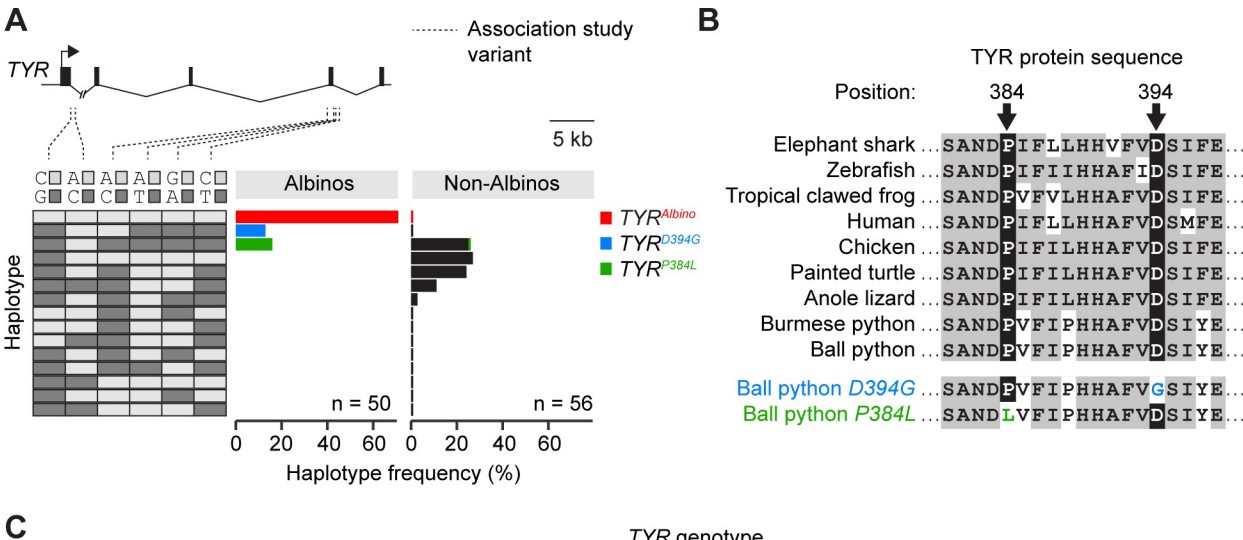

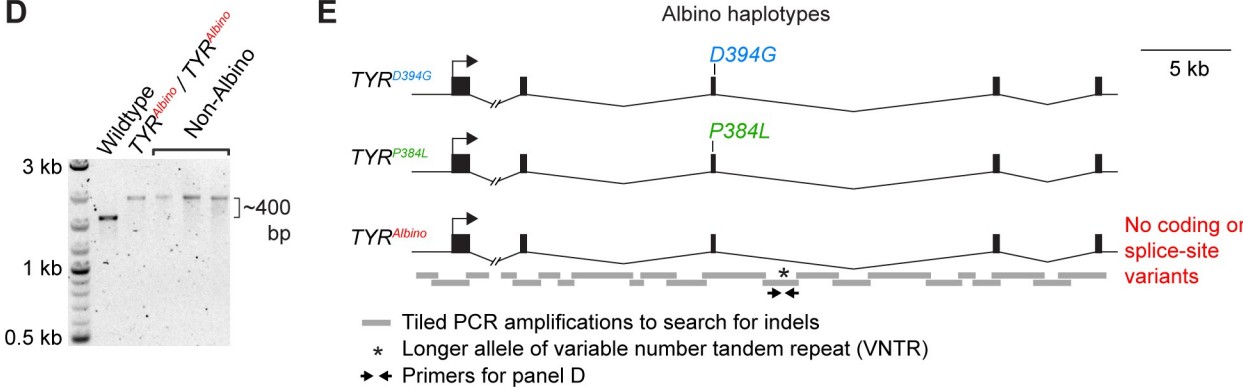

**Fig 2. The Albino color morph is associated with three haplotypes of *TYR*.** (A) *TYR* haplotype frequencies among Albinos and Non-Albinos. (B) Alignment of *TYR* protein sequences surrounding the missense variants *D394G* and *P384L*. (C) *TYR* genotypes and breeder designations of animals used in the association study. +, any *TYR* haplotype found exclusively among Non-Albinos. (D) PCR amplification of a genomic fragment containing the variable number tandem repeat (VNTR) shown in E. Non-Albinos are examples of Non-Albinos homozygous for the longer allele of the VNTR. (E) Schematic of the three *TYR* haplotypes found in Albinos. The *TYR^Albino^* haplotype contains no coding variants and no splice-site variants compared to wildtype. (A, E) Hash mark, discontinuity in the Burmese python reference genome.

## The *TYR^Albino^* haplotype lacks an obvious loss-of-function variant

The lack of coding or splice-site variants on the *TYR^Albino^* haplotype led us to hypothesize that this haplotype carried a loss-of-function variant in a non-coding region. To search for such variants, we examined the *TYR* promoter. We sequenced ~2 kb immediately upstream of the

*TYR* start codon in a $TYR^{Albino}$ homozygote and compared this sequence to wildtype. We found that the $TYR^{Albino}$ haplotype differed from wildtype at a total three sites (three single-base substitutions). In all three cases, the $TYR^{Albino}$ haplotype was homozygous for the allele shared with Burmese python (i.e. the ancestral allele). This result suggests that these variants are not causative for the Albino color morph. We conclude that the causative variant in the $TYR^{Albino}$ haplotype does not reside within the sequenced region, ~2 kb upstream of the *TYR* start codon.

We hypothesized that the $TYR^{Albino}$ haplotype might contain a large insertion or deletion (indel) in an intron that disrupts gene function. Intronic indels can disrupt slicing and have been found to disrupt the function of melanogenesis genes in other species [e.g. 62,74]. To search for large intronic indels, we tiled PCR amplicons across *TYR* introns and compared amplicon sizes between a $TYR^{Albino}$ homozygote and a wildtype animal. These amplicons tiled across a total of ~38 kb of intronic sequence, which included part of intron 1 and all of introns 2, 3, and 4 (Fig 2E). (Tiling across intron 1 was incomplete because this intron contains a discontinuity in the Burmese python reference genome. Amplification across this discontinuity was unsuccessful in ball python, even in wildtype.) Size differences between the $TYR^{Albino}$ homozygote and wildtype were assessed via standard agarose gel electrophoresis, which we estimate was sensitive enough to reveal indels larger than ~50–100 bp, depending on amplicon size (which ranged from ~0.7–3.5 kb). This analysis revealed one size difference (Fig 2D). Further sequencing revealed that this size difference was caused by a variable number tandem repeat (VNTR) that was ~400 bp larger in the $TYR^{Albino}$ homozygote than in wildtype. Genotyping of the 56 Non-Albinos for this indel revealed that eight of these animals were homozygous for the larger allele of the VNTR (Fig 2D). This finding suggests that the larger allele of the VNTR is not causative for the Albino color morph. We propose that the $TYR^{Albino}$ haplotype harbors a loss-of-function variant other than a large intronic indel or outside the genomic regions analyzed here.

## $TYR^{P384L}$ is associated with reduced phenotype severity compared to $TYR^{D394G}$ and $TYR^{Albino}$

Ball python breeders describe the Albino color morph as having three alleles: $Alb^{Albino}$, $Alb^{Candy}$, and $Alb^{Toffee}$. Evidence for this view comes in part from variation in the coloration of Albino animals. In some Albino animals, the brown-to-black coloration observed in wildtype is entirely absent, and skin patches appear white (Fig 1D). These Albinos are typically considered to be $Alb^{Albino}$ homozygotes. Other Albinos show a less severe phenotype, where skin patches are faintly beige or lavender instead of white (Fig 1E and 1F). These Albinos are typically considered to carry one or more copies of $Alb^{Candy}$ or $Alb^{Toffee}$.

We hypothesized that the Albino alleles recognized by breeders might correspond to the *TYR* haplotypes identified through sequencing ($TYR^{D394G}$, $TYR^{P384L}$, and $TYR^{Albino}$). To test this hypothesis, we examined the breeder designations of the 50 Albinos in our panel. We found that the $Alb^{Candy}$ or $Alb^{Toffee}$ designations typically corresponded the $TYR^{P384L}$ haplotype (Fig 2C). The $Alb^{Albino}$ designation typically corresponded to the other two haplotypes ($TYR^{D394G}$ and $TYR^{Albino}$, Fig 2C). This correspondence was imperfect, and exceptions existed (e.g. three animals designated as $Alb^{Albino}$ / $Alb^{Candy}$ compound heterozygotes did not carry the $TYR^{P384L}$ haplotype, Fig 2C). We conclude that the $Alb^{Candy}$ and $Alb^{Toffee}$ designations typically (but not exclusively) represent the $TYR^{P384L}$ haplotype. The $Alb^{Albino}$ designation typically (but not exclusively) represents the $TYR^{D394G}$ or $TYR^{Albino}$ haplotype. The association between the $Alb^{Candy}$ and $Alb^{Toffee}$ designations and the $TYR^{P384L}$ haplotype suggests that this haplotype may confer a slightly less severe phenotype than do $TYR^{D394G}$ and $TYR^{Albino}$.

## The Lavender Albino color morph is associated with a deletion in *OCA2*

The Lavender Albino color morph is characterized by skin patches that are lavender instead of brown or black. This phenotype is thought to arise from melanin levels that are dramatically reduced but not entirely eliminated. We hypothesized that this phenotype was caused by loss of function of *OCA2*, which encodes a chloride channel required for maintaining the pH of melanosomes [88,95]. When the OCA2 protein is absent or non-functional, the enzymes that synthesize melanin are less active, and only small amounts of melanin are produced [96,97]. The resulting phenotype is typically intermediate in severity between the loss-of-function phenotypes of *TYR* versus other melanogenesis genes (Table 1). *OCA2* was therefore a good candidate for the causative gene of the Lavender Albino color morph.

To search for loss-of-function variants in *OCA2*, we selected a single Lavender Albino animal and sequenced 23 of the 24 coding regions of *OCA2* in this animal. Comparison of these sequences to wildtype revealed no coding variants and no splice-site variants. We attempted to repeat this analysis for the remaining coding region (coding region 18), but we were unable to amplify this coding region from the Lavender Albino animal. Further test amplifications revealed that the Lavender Albino animal was homozygous for a 1,514-bp deletion spanning coding region 18 (Fig 3A and 3B). This deletion removes 36 amino acids from the protein and likely introduces a frameshift into the transcript, given that the coding regions downstream of the deletion are out of frame compared to coding regions upstream of the deletion (Fig 3E). This frameshifted transcript is predicted to produce a protein lacking six of the 12 transmembrane helices present in wildtype OCA2. Truncations occurring at similar positions in the OCA2 protein have been identified in humans and are associated with oculocutaneous albinism [98]. We conclude that the deletion of *OCA2* coding region 18 likely disrupts protein function and is a strong candidate for the cause of the Lavender Albino color morph.

The Lavender Albino color morph is considered by breeders to have a single allele. We therefore predicted that the *OCA2* deletion would be shared by other Lavender Albino animals. We predicted that Non-Lavender Albinos would be heterozygous for the deletion or non-carriers. To test this prediction, we genotyped the *OCA2* deletion in 13 additional Lavender Albino animals. We also genotyped 76 Non-Lavender Albinos and one animal described as heterozygous for the Lavender Albino color morph. We found that all 13 Lavender Albinos were homozygous for the deletion (Fig 3C and 3D). All 76 Non-Lavender Albinos were non-carriers. The animal reported to be heterozygous for Lavender Albino was heterozygous for the deletion. These findings support the conclusion that the *OCA2* deletion is causative for the Lavender Albino color morph (Fig 3E).

## The Ultramel color morph is associated with a missense variant and a putative deletion in *TYRP1*

The Ultramel color morph is characterized by skin patches that are tan or light brown, rather than dark brown or black. This phenotype suggests a mild loss of melanin. We hypothesized that this phenotype was caused by loss of function of one of five genes: *TYRP1*, *TYRP2*, *SLC7A11*, *SLC24A5*, and *SLC45A2* (Table 1). *TYRP1* and *TYRP2* encode enzymes involved in synthesizing melanin [87]. Their loss-of-function phenotypes are mild because of partially redundancy with other enzymes in the melanin synthesis pathway. *SLC7A11* encodes a transporter responsible for importing cystine into the cell [99]. Cystine is a precursor to some forms of melanin, and its reduction alters melanin levels [77]. *SLC24A5* encodes a $K^+$-dependent $Na^+$-$Ca^{2+}$ exchanger [89]. *SLC45A2* encodes a putative sugar transporter [90]. Loss of their encoded proteins reduces melanin through mechanisms that may involve defects in

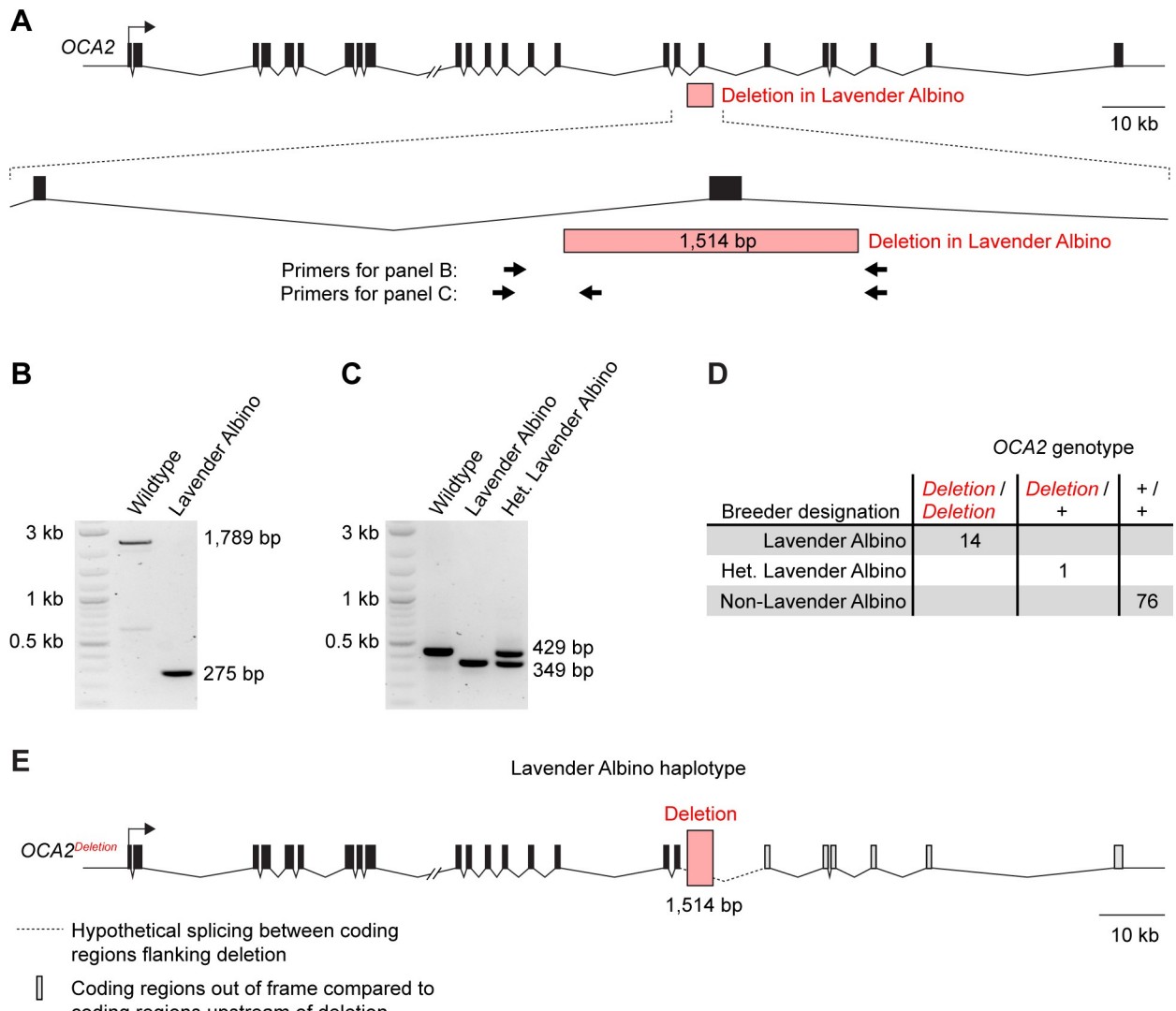

**Fig 3. The Lavender Albino color morph is associated with a deletion in *OCA2*.** (A) Schematic of the *OCA2* gene. (B) PCR amplification demonstrating the deletion in Lavender Albino. (C) PCR amplification used for genotyping the deletion in the animals in panel D. (D) Genotypes of 14 Lavender Albinos, 76 Non-Lavender Albinos, and one animal described as heterozygous for Lavender Albino. This set of animals includes the original Lavender Albino animal in which the OCA2 deletion was identified. (E) Schematic of the *OCA2* haplotype found in Lavender Albinos. (A, E) Hash mark, discontinuity in the Burmese python reference genome.

regulation of melanosome pH [84,89,100]. These fives genes were therefore good candidates for the causative gene of the Ultramel color morph.

To search for loss-of-function variants in *TYRP1*, *TYRP2*, *SLC7A11*, *SLC24A5*, and *SLC45A2*, we selected a single Ultramel animal and sequenced the coding regions of each of these genes in this animal. Comparison of these sequences to wildtype revealed a single coding variant: a missense variant in the fourth coding region of *TYRP1* (MZ269497:c.G1720A), which leads to an arginine-to-histidine exchange [UPQ41334.1:p.Arg305His]. The Ultramel animal was homozygous for this variant, termed hereafter *R305H*. The arginine residue at this site is conserved across vertebrates (Fig 4A) and is also conserved in *TYR*, which is paralogous to *TYRP1* [101]. An arginine-to-histidine substitution at the homologous site in *TYR* has been reported in humans and is associated with oculocutaneous albinism [102–104]; thus, histidine

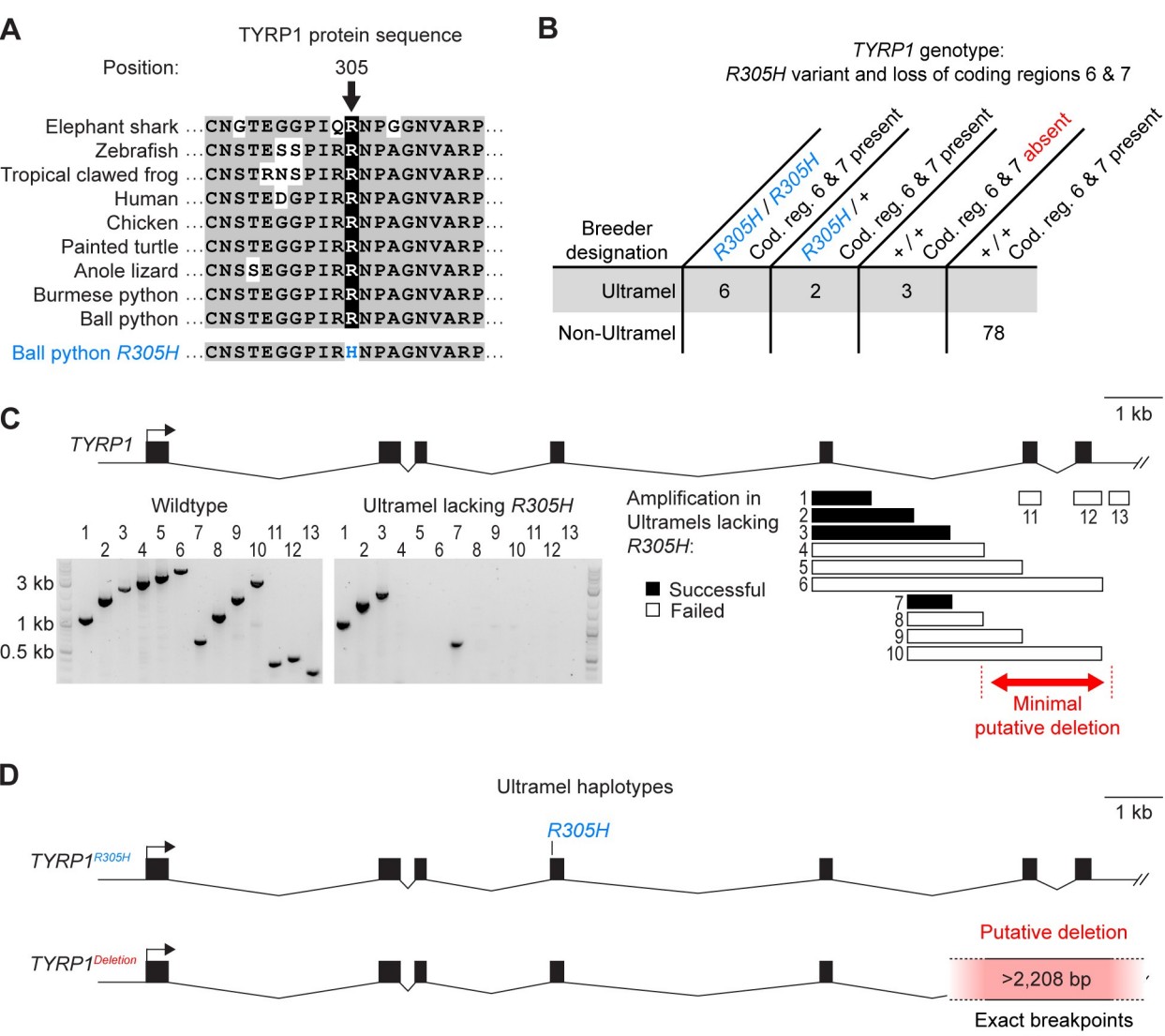

**Fig 4. The Ultramel color morph is associated with a missense variant and a putative deletion in *TYRP1*.** (A) Alignment of *TYRP1* protein sequence surrounding missense variant *R305H*. (B) Genotypes of 11 Ultramels and 78 Non-Ultramels. This set of animals includes the original Ultramel animal in which the *R305H* variant was identified. Ultramels heterozygous for *R305H* are presumed to be heterozygous for the putative deletion of coding regions 6 and 7; testing these animals for the putative deletion was not possible because the putative deletion cannot be detected when heterozygous. (C) Top, schematic of the *TYRP1* gene. Bottom left, PCR amplifications demonstrating the putative deletion in the Ultramels lacking *R305H*. Bottom right, alignment of PCR amplicons to the *TYRP1* gene. (D) Schematic of the two *TYRP1* haplotypes found in Ultramels. (A, D) Hash mark, discontinuity in the Burmese python reference genome.

at this site is likely disruptive to protein function. The *R305H* variant is therefore a good candidate for the cause of the Ultramel color morph.

The Ultramel color morph is considered by breeders to have a single allele. We therefore predicted that the *R305H* variant would be shared by other Ultramels. We predicted that Non-Ultramels would be heterozygous or non-carriers. To test this prediction, we genotyped the *R305H* variant in 10 additional Ultramels and 78 Non-Ultramels. We found that five of the 10 Ultramels were homozygous for the *R305H* variant (Fig 4B). Of the remaining Ultramels, two were heterozygous for the *R305H* variant, and three did not carry it. By contrast, none of the 78 Non-Ultramels carried the *R305H* variant. This pattern is consistent with the *R305H* variant causing the Ultramel phenotype in some Ultramels but not others.

We hypothesized that the *R305H* variant represented one of two loss-of-function alleles of *TYRP1*. Under this scenario, Ultramels lacking the *R305H* variant are predicted to be homozygous for an alternate loss-of-function allele. Ultramels heterozygous for *R305H* are predicted to be compound heterozygotes. To test this prediction, we amplified and sequenced coding regions 1 through 5 of *TYRP1* in one of the three Ultramels lacking the *R305H* variant. We found no coding variants and no splice-site variants compared to wildtype. We attempted to repeat this analysis for coding regions 6 and 7 of *TYRP1*, but we were unable to amplify these coding regions from this animal (Fig 4C). Identical results were observed for the other two Ultramels lacking the *R305H* variant (S5 Table). Further test amplifications indicated that coding regions 6 and 7 were missing from the genomes of all three animals (Fig 4C, S5 Table). Loss of these coding regions was specific to Ultramels lacking the *R305H* variant and did not occur in any of the Non-Ultramels, nor in the Ultramels homozygous or heterozygous for *R305H* (Fig 4B). A simple explanation of this pattern is that the Ultramels lacking *R305H* were homozygous for a deletion spanning coding regions 6 and 7 of *TYRP1*. The two Ultramels heterozygous for *R305H* were presumed to be compound heterozygous for this putative deletion, although we were not able to confirm this heterozygosity because the putative deletion extended into a discontinuity in the Burmese python reference genome (and therefore could not be detected in heterozygotes). The putative deletion removes at least 2,208 bp from the genome and 117 amino acids from the TYRP1 protein, including the second of two zinc-binding domains. Truncations occurring at similar positions in the TYRP1 protein have been identified in humans and are associated with oculocutaneous albinism [105]. We conclude that the Ultramel color morph is likely caused by variants in *TYRP1* and has two alleles: missense variant *R305H* and a putative deletion of coding regions 6 and 7 (Fig 4D).

## Discussion

The goal of our study was to use pet samples recruited from the community to identify the genetic causes of the Albino, Lavender Albino, and Ultramel color morphs. We succeeded in recruiting 11 or more animals for each morph, along with a larger number of animals not belonging to these morphs. This sample size, albeit small, was sufficient to identify putatively causal variants for each morph (Fig 5). The Albino color morph was associated with three alleles of TYR: missense variant *D394G*, missense variant *P384L*, and haplotype $TYR^{Albino}$, which lacks coding or splice-site variants compared to wildtype. The Lavender Albino color morph was associated with a single allele of *OCA2*, a 1,514-bp deletion that removes *OCA2* coding region 18. The Ultramel color morph was associated with two alleles of *TYRP1*: missense variant *R305H* and a putative deletion that removes *TYRP1* coding regions 6 and 7. Due to the small sample size of our study, we cannot exclude the possibility of additional loss-of-function alleles of these genes segregating in the ball python population. However, such alleles are expected to be at low frequency, given their absence in our sample. These findings are consistent with genetic data from other vertebrates (Table 1), indicating that the loss-of-function phenotypes of these genes range from severe (*TYR*) to moderate (*OCA2*) to mild (*TYPR1*). Our study demonstrates that pet ball pythons are a tractable resource for genetic analysis of coloration in reptiles, at least for color morphs having obvious candidate genes.

### Molecular functions of genetic variants associated with color morphs

The missense variants and deletions found in *TYR*, *TYRP1*, and *OCA2* are likely hypomorphic or null alleles. The deletion in *OCA2* likely results in a frameshifted transcript that, if not degraded by nonsense-mediated decay, encodes a protein lacking six of the 12 transmembrane helices present in wildtype OCA2 [95]. This truncated protein is therefore unlikely to retain the Cl⁻ channel activity observed for wildtype OCA2 [88]. *TYR* and *TYPR1* encode globular

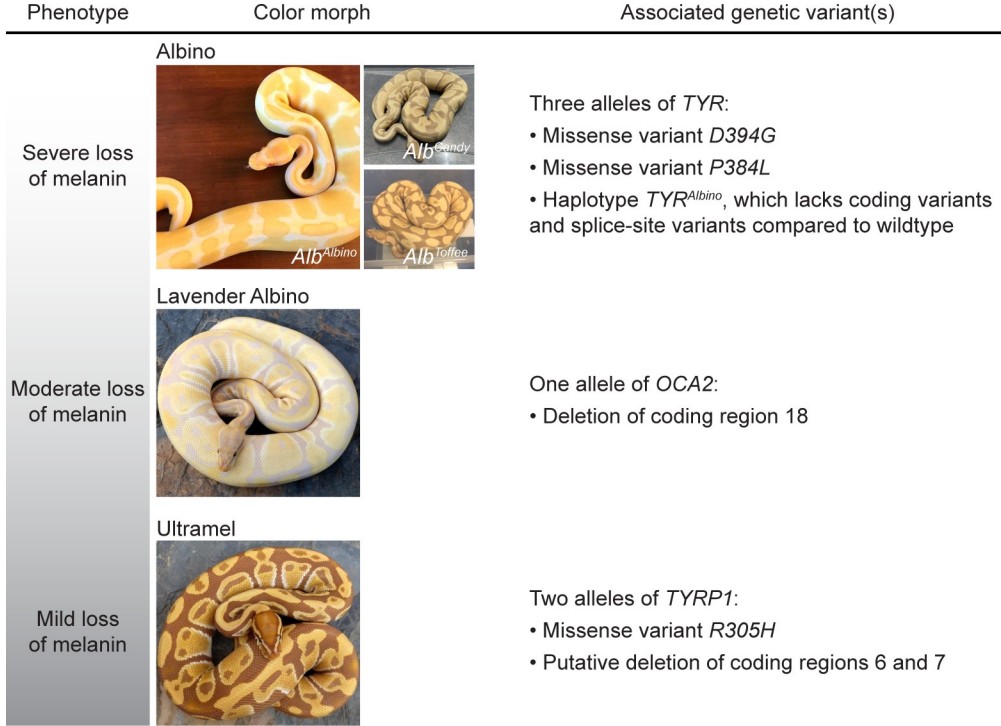

| Phenotype | Color morph | Associated genetic variant(s) |
|---|---|---|

**Fig 5. Summary of genetic variants associated with the Albino, Lavender Albino, and Ultramel color morphs.** Photo credits, Ryan Young of Molecular Reptile, Chiron Graves, Phil Barclay, Michael Freedman of The Florida Reptile Ranch.

enzymes that require copper or zinc as co-factors [87]. Variants *P384L* and *D394G* in *TYR* reside in the second of TYR's two copper-binding domains and may therefore alter the ability of the TYR to bind copper. Variant *R305H* in *TYRP1* occurs at a site that normally forms salt bridges with residues located elsewhere in the peptide chain [87]. An arginine-to-histidine substitution at this site likely disrupts these salt bridges and may therefore interfere with proper protein folding. The putative deletion in *TYRP1* removes the C-terminal ~20% of the protein. This deleted region includes the second of TYRP1's two zinc-binding domains and several alpha helices residing to the enzyme's hydrophobic core [87]. Truncated TYRP1 protein is therefore unlikely to fold properly, nor is it likely to properly associate with zinc.

The *TYR^Albino* haplotype is the most common *TYR* haplotype found among Albinos, but it lacks an obvious loss-of-function variant. This haplotype contains no coding variants, no splice-site variants, no derived variants within 2 kb upstream of the start codon, and no large indels in three of four introns (with the exception of a longer allele of a VNTR, which is not specific to the *TYR^Albino* haplotype). We propose that the *TYR^Albino* haplotype contains a loss-of-function variant not detectable by methods used in the current study. Examples include regulatory variants farther upstream of the start codon, substitutions in introns that disrupt splicing, or large indels or rearrangements involving the first intron. Cryptic loss-of-function variants in melanogenesis genes are thought to be relatively common in humans, where ~10–20% of oculocutaneous albinism patients are heterozygous for pathogenic variants in a known melanogenesis gene but lack a coding or splice-site variant on the opposite allele [106–108].

## Multiple alleles of the Albino color morph

Prior to the current study, the Albino color morph was recognized by breeders as having three alleles: *Alb^Albino*, *Alb^Candy*, and *Alb^Toffee*. *Alb^Albino* was thought to confer a more severe

phenotype than $Alb^{Candy}$ and $Alb^{Toffee}$ (Fig 1D–1F). Our results confirm that the Albino color morph has three molecular alleles ($TYR^{D394G}$, $TYR^{P384L}$, and $TYR^{Albino}$). Yet these alleles do not perfectly correspond to the alleles recognized by breeders. The molecular allele $TYR^{P384L}$ typically corresponds to the breeder designations $Alb^{Candy}$ and $Alb^{Toffee}$. The molecular alleles $TYR^{D394G}$ and $TYR^{Albino}$ typically correspond to the breeder designation $Alb^{Albino}$ (Fig 2C). This correspondence is imperfect and exceptions exist (Fig 2C).

We speculate that the $TYR^{P384L}$ haplotype was discovered twice and was named $Alb^{Candy}$ by one breeder and $Alb^{Toffee}$ by another breeder. We speculate that $TYR^{D394G}$ and $TYR^{Albino}$ confer similar phenotypes and were not previously recognized by breeders as distinct. The less severe phenotype associated with $TYR^{P384L}$ may reflect the $P384L$ variant being less disruptive to gene function compared with variants on the other two alleles. Alternatively, the less severe phenotype associated with $TYR^{P384L}$ may reflect genetic linkage to modifiers in other genes. In either case, our study demonstrates that the Albino allele designations current in use by breeders do not accurately reflect molecular genotype. Renaming of the Albino alleles is warranted, although owners and breeders may be resistant to renaming due to cultural attachment to existing allele names.

## Multiple alleles of the Ultramel color morph

Our finding of multiple *TYRP1* alleles for the Ultramel color morph was unexpected. This morph was not previously described as having multiple alleles. We speculate that one of the two alleles associated with the Ultramel color morph may represent an allele originally associated with a morph known as Caramel Albino. Multiple lineages of Caramel Albinos have been described, and their coloration similar to Ultramels. Caramel Albinos have been disfavored among owners and breeders because of spinal kinking and reduced female fertility. Caramel Albinos were not included in our study because we were unable to recruit any Caramel Albino samples. We speculate that the Caramel Albino morph may be allelic with Ultramel, and that one of two *TYRP1* alleles associated with Ultramel may represent an allele originally described as Caramel Albino. Alternately, Ultramel may be distinct from Caramel Albino, and the two *TYRP1* alleles may represent two distinct origins of the Ultramel morph.

## Prospects for genetic testing

Many breeders of ball pythons wish to identify heterozygotes of recessive color morphs. Currently the only tool for identifying heterozygotes is test breeding, which is slow and laborious. The results of the current study will enable simple genetic testing for Albino, Lavender Albino, and Ultramel, which will allow heterozygotes to be identified prior to reproductive maturity. Testing for Lavender Albino can be performed by genotyping the *OCA2* deletion. Testing for Albino can be performed by sequencing the *TYR* missense variants *D394G* and *P384L*, and by genotyping variants that distinguish the $TYR^{Albino}$ haplotype from other *TYR* haplotypes. Detection of *D394G* or *P384L* can be considered diagnostic for Albino because these variants are likely causative for the Albino color morph. Detection of the $TYR^{Albino}$ haplotype is less diagnostic because the causative variant on $TYR^{Albino}$ haplotype remains unknown. Testing for one of the two Ultramel alleles can be performed by sequencing the *TYRP1* missense variant *R305H*. The other Ultramel allele—the putative deletion of *TYRP1* coding regions 6 and 7—cannot currently be detected in heterozygotes using simple methods because the breakpoints of this putative deletion are unknown.

## Use of Burmese python reference genome

One challenge for genetic studies in ball python is the absence of a reference genome. To fill this gap, we relied on the genome of Burmese python [91]. This genome is a scaffold-level

assembly, and some genes are fragmented across scaffolds. This fragmentation limited our analysis of the $TYR^{Albino}$ haplotype and the putative deletion in *TYRP1*. For $TYR^{Albino}$, a discontinuity in the first intron of *TYR* in the Burmese python genome prevented us from assessing whether the $TYR^{Albino}$ haplotype contained a large indel or rearrangement in this intron. For the putative deletion in *TYRP1*, a discontinuity immediately downstream of *TYRP1* prevented us from mapping the exact breakpoints of the deletion. As genomic resources for Burmese python and ball python expand, ideally with the use long-read technologies to improve genome assemblies and identify structural variants, we expect to identify a putatively causative variant on the $TYR^{Albino}$ haplotype and to map the breakpoints of the putative deletion in *TYRP1*. This information will increase the prospects for genetic testing for these alleles.

### Ball pythons as a genetic system

The market for pet ball pythons is huge, and many owners prefer pets with novel color patterns. This demand has led breeders to propagate genetic variants affecting coloration [31,33]. This effort has fortuitously created a collection of "mutants" useful for understanding the genetics of coloration in reptiles. Examples include ball python morphs with altered red-to-yellow coloration (e.g. Axanthic) and morphs in which the normal mottled color pattern is converted to dorsal stripes (e.g. Clown, Genetic Stripe, Super Stripe). Red-to-yellow pigments are largely uncharacterized in reptiles [although see 28,29], and morphs affecting these pigments may provide insight into the metabolism and storage of these pigments. Morphs with stripes are reminiscent of evolutionary changes in color patterning across snake species [109] and may provide insights into the developmental mechanisms by which these changes evolve. Future studies of ball python color morphs will be aided by our groundwork showing that ball python samples can be recruited effectively from pet owners, and that genetic analyses in ball python can be scaffolded using the genome of Burmese python as a reference. Similar groundwork has also been provided by a recent study characterizing the ball python morph known as Piebald [26]. We expect that a continued community-science approach will be effective in developing ball pythons into a resource for understanding the genetics of reptile coloration.

## Materials and methods

### Recruitment of ball python sheds

Ball python sheds were recruited from pet owners and breeders by placing announcements in Twitter, Reddit, Instagram, and Facebook, and by contacting sellers having active listings on Morph Market (www.morphmarket.com). Contributors were instructed to allow sheds to dry (if wet) and to ship sheds via standard first-class mail. Contributors sending multiple sheds were instructed to package sheds individually in plastic bags during shipping. Contributors were not provided monetary compensation for sheds, although some contributors were given pre-paid shipping envelopes to cover shipping costs. Contributors were thanked via social media whenever possible. Upon receipt, sheds were stored at -20˚C to kill any insect larvae infesting the sheds.

To maximize genetic diversity within each category of morph, we limited our sample of animals of any one morph to animals contributed by different contributors. Exceptions were made if animals contributed by the same contributor had been obtained from different breeders. The goal of this strategy was to reduce the number of animals that were close relatives (e.g. siblings or parent-offspring pairs). Obtaining full pedigree information was not possible because most owners lacked this information (e.g. "I bought my animal at a pet store" or "I got my animal from a friend who was moving away"). The idea that animals in our sample were derived from multiple lineages is supported by our discovery of multiple alleles for the Albino

and Ultramel color morphs. However, we cannot exclude the possibility that some animals in our sample may have been close relatives.

The total set of animals comprised 50 Albinos, 14 Lavender Albinos, 11 Ultramels, one animal described as heterozygous for Lavender Albino, and 46 animals described as having normal coloration (i.e. wildtype) or belonging to color morphs other than Albino, Lavender Albino, or Ultramel. The Albinos were composed of 36 animals described as $Alb^{Albino}$ homozygotes, three animals described as $Alb^{Candy}$ homozygotes, one animal described as an $Alb^{Toffee}$ homozygote, nine animals described as $Alb^{Albino}$ / $Alb^{Candy}$ compound homozygotes, and one animal described as an $Alb^{Albino}$ / $Alb^{Toffee}$ compound homozygote. We use the term Non-Albino to refer to animals having normal coloration or belonging to morphs other than Albino. The Non-Albinos included some Lavender Albinos and some Ultramels. The terms Non-Lavender Albino and Non-Ultramel were used in analogous ways.

Phenotypes of Albinos, Lavender Albinos, and Ultramels were confirmed by examining shed skins for a reduction of brown-to-black coloration. The single animal described as heterozygous for Lavender Albino was included to demonstrate the three-primer assay for genotyping the OCA2 deletion. Heterozygotes for Albino or Ultramel and additional heterozygous for Lavender Albino were excluded from our study because we have found that animals described as heterozygous for recessive traits by breeders and owners are sometimes not heterozygous for these traits (e.g. due to mis-attribution of paternity).

## Performing experiments in an undergraduate laboratory course

The majority of experiments and analyses described in this study were performed by undergraduate students as part of a laboratory course at Eastern Michigan University (BIO306W). This practice required that our experimental design rely on simple techniques, namely PCR and Sanger sequencing. To avoid student errors in these techniques, we implemented the following precautions. First, students never extracted DNA from more than one animal within the same laboratory period. Second, students performed negative and positive control reactions for all PCR amplifications. Data from students having incorrect controls were excluded from analysis. Third, all sequence analyses were performed independently by three or more students. When the results of all students did not all agree, sequences were re-analyzed by the instructor (HSS).

## Annotation of melanogenesis genes in Burmese python

Our analyses in ball python used the genome of Burmese python as a reference. We therefore required high-confidence gene annotations in Burmese python for the genes used in our study (*TYR*, *TYRP1*, *TYRP2*, *OCA2*, *SLC7A11*, *SLC24A5*, and *SLC45A2*). Preliminary inspection of existing gene annotations in Burmese python suggested that many gene annotations contained errors (e.g. missing one or more coding regions). A main cause of these errors was that the Burmese python genome (*Python_molurus_bivittatus-5.0.2*) is a scaffold-level assembly, and many genes were split across scaffolds. We therefore curated new gene annotations in Burmese python, using conservation of gene structure across species. Alignment of protein sequences for each gene from corn snake, anole lizard, chicken, and/or mouse revealed that gene structure was highly conserved for all seven genes: Genes in each species contained the same number of coding regions, and the coding-region boundaries relative to protein sequences were perfectly conserved across species, with the exception of one slight difference in mouse for one boundary of *SLC45A2* (S2 Table). We therefore felt confident that conservation of coding-region boundaries could be used to curate new gene annotations in Burmese python. While it remains theoretically possible that our new gene annotations contain minor errors, our

confidence in these gene annotations is high, given the high conservation of gene structure across species.

To curate new gene annotations, we performed TBLASTN [110,111] searches of the Burmese python genome (*Python_molurus_bivittatus-5.0.2*) using protein sequences from anole lizard or corn snake as the query. TBLASTN is a search tool that aligns a query protein sequence all six reading frames of a nucleotide database. TBLASTN searches were performed using default parameters. Hits were examined manually. Gene annotations were built to match the coding-region boundaries conserved across species. Details for each gene annotation are given below.

**TYR.** The Burmese python genome was queried using TYR protein sequence from anole lizard (*XP_003219419.1*). The N-terminus of this query (coding region 1) hit Burmese python transcript *XM_007438960.1* on scaffold 4418. The C-terminus of this query (coding regions 2 to 5) hit Burmese python transcript *XM_007436041.2* on scaffold 3103. TYR was annotated as the union of these transcripts. The 3' boundary of coding region 1 was adjusted to match the boundary conserved across species (...TTC TCT TCA TGG CAA−3').

**TYRP1.** The Burmese python genome was queried using TYRP1 protein sequence from corn snake (*XP_034266320.1*). The N-terminus of this query (coding regions 1 to 5) hit Burmese python transcript *XM_007426971.3* on scaffold 801. The C-terminus of this query (coding regions 6 and 7) hit unannotated regions on this same scaffold. *TYRP1* was annotated as the union of transcript *XM_007426971.3* and coding regions 6 and 7. The boundaries of coding regions 6 and 7 were built to match the boundaries conserved across species (coding region 6, 5'-ATA TCT CAA CAT ACC....AAG TTC AGT GGC CAT-3'; coding region 7, 5'-CAC AAG CTC TCC ATG....CAG TCA GAT GTG TGA-3').

**TYRP2.** The Burmese python genome was queried using TYRP2 protein sequence from corn snake (*XP_034273310.1*). The N-terminus of this query (coding regions 1 to 3) hit unannotated regions on Burmese python scaffold 4970. The C-terminus of this query (coding regions 4 to 8) hit Burmese python transcript *XM_025174219.1* on this same scaffold. *TYRP2* was annotated as the union of coding regions 1 to 3 and C-terminal five coding regions of transcript *XM_025174219.1*. Transcript *XM_025174219.1* contains a sixth coding region that does not match *TYRP2* and was therefore excluded from the gene annotation. The boundaries of coding regions 1 to 3 were built to match the boundaries conserved across species (coding region 1, 5'-ATG GCC TTC CTG CTG....GTT GCC AAT GCA CAG-3'; coding region 2, 5'-GAC ATT TTG CTG GCT....GAG ATA CTC TAT TAG-3'; coding region 3, 5'- GAC CAG GCC GTC CCT....GAA AGA GAT CTG CAG-3').

**OCA2.** The Burmese python genome was queried using OCA2 protein sequence from corn snake (*XP_034287267.1*). The N-terminus of this query (coding regions 1 to 9) hit Burmese python transcript *XM_025173964.1* on scaffold 4704. The C-terminus of this query (coding regions 10 to 24) hit Burmese python transcript *XM_007433276.1* on scaffold 2194. *OCA2* was annotated as the union of these transcripts. The 5' boundary of coding region 10 was adjusted to match the boundary conserved across species (5'-ATT GTC CAC AGG ACA...)

**SLC7A11.** The Burmese python genome was queried using SLC7A11 protein sequence from corn snake (*XP_034257397.1*). All regions of the query hit Burmese python transcript *XM_007430906.3* on scaffold 1543. *SLC7A11* was annotated as transcript *XM_007430906.3*, with no further adjustments.

**SLC24A5.** The Burmese python genome was queried using SLC24A5 protein sequence from corn snake (*XP_034290605.1*) extended at its N-terminus to an upstream start codon present in the parent transcript (*XM_034434714.1*). The N-terminus of the query (coding region 1) hit an unannotated region on Burmese python scaffold 3984. The C-terminus of this query (coding regions 2 to 9) hit Burmese python transcript *XM_007438007.2* on this same

scaffold. *SLC24A5* was annotated as the union of coding region 1 and transcript *XM_007438007.2*. The boundaries of coding region 1 were built to match the boundaries conserved across species (5'-ATG CAG CCT GCC GAG....TCC GCG AGG ATC CCG -3'). The 5' boundary of coding region 2 was adjusted to match the boundary conserved across species (5'-AGA ACG AAA CCC GCT...).

**SLC45A2.** The Burmese python genome was queried using *SLC45A2* protein sequence from corn snake (*XP_034298386.1*). The N-terminal region of this query (coding regions 1 and 2) hit Burmese python transcript *XM_007432459.2* on scaffold 1939. The C-terminal region of this query (coding regions 5 to 7) hit Burmese python transcript *XM_007437721.2* on scaffold 3858. *SLC45A2* was annotated as the union of these transcripts.

## DNA extraction

Sheds were rinsed in tap water to remove dirt and debris. Sheds were air dried and lysed overnight at ~60˚C in ~1 ml lysis buffer (100 mM Tris-HCl pH 8.0, 100 mM EDTA, 2% sodium dodecyl sulfate, 3 mM $CaCl_2$, 2 mg/ml Proteinase K) per ~8 $cm^2$ piece of shed. Lysate was separated from visible fragments of undigested shed and further cleared by centrifugation at 13,000 x g for 2 min. To precipitate protein, ammonium acetate was added to supernatant to a final concentration of 1.875 M. Samples were incubated on ice for 5–10 min and centrifuged at 13,000 x g for 3–5 min at 4˚C. Supernatant was mixed with an equal volume of magnetic bead mixture (10 mM Tris-HCl pH 8.0, 1 mM EDTA, 1.6 M NaCl, 0.2% Tween-20, 11% polyethylene glycol, 0.04% washed SpeedBeads [Sigma #GE45152105050250]), and samples shaken for 5–10 min. Beads were separated from supernatant using a magnet, washed twice in 0.2 ml 70% ethanol for 2 min, and air dried for ~1 min. DNA was eluted from beads in TE buffer (10 mM Tris-HCl pH 8.0, 1 mM EDTA) at 65˚C for >5 min.

## Primer design and PCR

Primers were designed against the genome of Burmese python or against genomic sequences from ball python obtained in an earlier step of the study. Primers were designed using Primer3 [112], using default parameters and a target annealing temperature of 60˚C. Amplification was first tested at 57˚C, to allow for occasional divergence between ball python and Burmese python genomic sequences. In some cases, annealing temperatures were later adjusted to 52˚C, 60˚C, or 61˚C, to obtain stronger product or to eliminate background bands.

Genomic fragments were amplified using OneTaq polymerase (NEB #M0480) or Q5 polymerase (NEB #M0491). Genotyping assays described below used OneTaq, unless otherwise specified. OneTaq reactions consisted of 1X OneTaq Standard Reaction Buffer, 200 μM dNTPs, 0.2 μM of each primer, and 0.025 U/μl OneTaq polymerase. OneTaq thermocycling conditions were as follows: 94˚C for 2 min; 30–35 cycles of 94˚C for 30 sec, 52–61˚C for 30 sec, and 68˚C for 1–4 min; and 68˚C for 5 min. Q5 reactions consisted of 1X Q5 Reaction Buffer, 200 μM dNTPs, 0.5 μM of each primer, and 0.02 U/μl Q5 polymerase. Q5 thermocycling conditions were as follows: 98˚C for 30 sec; 30–35 cycles of 98˚C for 10 sec, 58–61˚C for 15 sec, and 72˚C for 1.5–3 min; and 72˚C for 5 min. Reactions used 10–100 ng template DNA per 20 μl volume.

## Sanger sequencing

PCR products were purified for Sanger sequencing using magnetic beads or gel extraction. For magnetic-bead purification, PCR reactions were mixed with three volumes of magnetic-bead mixture (10 mM Tris-HCl pH 8.0, 1 mM EDTA, 1.6–2.5 M NaCl, 0.2% Tween-20, 11–20% polyethylene glycol, 0.04% washed SpeedBeads [Sigma #GE45152105050250]), and agitated

for 5 min. Beads were separated from supernatant using a magnet, washed twice in 0.2 ml 80% ethanol for >30 sec, and air-dried for 30 sec. PCR products were eluted from beads in 10 mM Tris-HCl pH 8.0 for >3 min at 65˚C. Gel extraction was performed using QIAquick Gel Extraction Kit (Qiagen #28704), according to the manufacturer guidelines. Sanger sequencing was performed by Eton Bioscience Inc (etonbio.com).

### Sequencing of coding regions and comparison to Burmese python

Coding regions of melanogenesis genes were amplified and sequenced using primers given in S3 Table. Chromatograms were trimmed using SnapGene Viewer (snapgene.com/snapgene-viewer) and aligned to one another or to genomic sequences from Burmese python using ApE (jorgensen.biology.utah.edu/wayned/ape). Alignments were examined manually to identify divergent and polymorphic sites. Sequence identity between ball python and Burmese python was calculated across alignable sequence, excluding indels.

### Association study genotyping

Variants for the association study were identified through amplification and pilot sequencing of genomic fragments from five animals (four Non-Albinos and one Albino). For *TYR*, *TYRP1*, and *TYRP2*, we identified one or more genomic fragments containing multiple polymorphic sites within the same amplicon. These genomic fragments were selected for the association study. For *OCA2*, we did not identify any genomic fragments containing multiple polymorphic sites; we therefore selected two genomic fragments each containing a single polymorphic site. Divergence between the Albino animal and the other four animals was not a criterion for inclusion of variants in the association study. Variants were genotyped by amplifying and sequencing genomic fragments containing the variants. Locations of variants and primers used for amplifying and sequencing variants are given in S1 Table. Genotypes are provided in S6 Table.

### Haplotype reconstruction

Haplotypes were reconstructed using PHASE version 2.1 [92,93]. Parameters were set to 200 iterations, a thinning interval of 2, and a burn-in parameter of 100. These settings produced identical or nearly identical output for seven runs seeded with different random numbers; thus, these settings met the criteria for effective choice of parameter settings, according to the PHASE documentation. The case-control permutation test was performed by comparing Albinos to Non-Albinos.

### Genotyping assays for *TYR*

*TYR* missense variants *D394G* and *P384L* were genotyped by amplifying and sequencing a genomic fragment containing *TYR* coding region 3. This fragment was amplified using primers 13F (5'-ACT TTC AGG TGG GCA GCA G-3') and 13R (5'-GCT GAC AAC TAA AAT CTC TGC AA-3') and an annealing temperature of 52˚C, or using primers 242F (5'-GCC ATT GTA GCT TCT TAC CAC TC-3') and 242R (5'-TTC CAG TCC ATA TAC AAG ATA TCC AA-3') and an annealing temperature of 57˚C. The amplicon was sequenced using primer 13F or 242F. Genotypes are provided in S6 Table.

*TYR* promoter and intronic regions are amplified using primers given in S4 Table. Promoter fragments were sequenced in full, and intronic fragments were sequenced from one end, to confirm that the correct region of the genome had been amplified. Sequencing primers are given in S4 Table. Size differences among intronic fragments were assessed by separating

fragments on a 1.25% agarose gel. Gels were run long enough for the shortest ladder band (100 bp) to migrate ~8 cm from its starting position.

Allele sizes of the variable number tandem repeat (VNTR) in *TYR* were genotyped by amplifying a genomic fragment located in *TYR* intron 3, using primers 214F (5'-TCT CAC CTG ATG GCA CAT TC-3') and 209R (5'-GTG CCC ACC CTG ATG TTA TT-3') and an annealing temperature of 60˚C. Amplicon sizes were analyzed as for intronic fragments, described above.

### Genotyping assays for *OCA2*

The *OCA2* deletion was initially identified by amplifying and sequencing a genomic fragment spanning *OCA2* coding region 18. This fragment was amplified using primers 218F (5'-ACC CCG TAG CCT CTT CAA AT-3') and 166R (5'-TGG GTG GCA AAC AAT CAT AA-3'), an annealing temperature of 60˚C, and Q5 polymerase. Amplicons were sequenced using both primers.

The *OCA2* deletion was genotyped after its initial identification using a three-primer PCR assay. This assayed used one forward primer and two reverse primers. The forward primer and one of the reverse primers were located outside the deletion: 217F (5'-GGA GAG AGA ATC CAA CCC TTG −3') and 166R (5'-TGG GTG GCA AAC AAT CAT AA-3'). The second reverse primer was located within the deletion: 188R (5'-CAA AGA CCA TTG TCC ATT TCC-3'). The annealing temperature was 57˚C. This assay produces a 429-bp product for the wildtype allele and a 349-bp product for the deletion allele. Heterozygotes produce both products. Genotypes are provided in S6 Table.

### Genotyping assays for *TYRP1*

*TYRP1* missense variant *R305H* was genotyped by amplifying and sequencing a genomic fragment spanning *TYRP1* coding region 4. This fragment was amplified using primers 18F (5'-GCT CTT TTC TCT AAG TCT GAC CTC -3') and 18R (5'-TCT TGT CCC ACA AAA GGA TTT -3') and an annealing temperature of 57˚C. The amplicon was sequenced using primer 18F.

The putative deletion of *TYRP1* coding regions 6 and 7 was identified using primers given in S5 Table. The putative deletion was genotyped after its initial identification by amplifying a genomic fragment of *TYRP1* spanning coding regions 6 and 7. This fragment was amplified using primers 20F (5'-GCA TTG TTT TAT CAG CCA TGA A-3') and 21R (5'-GGA ATT GAG ACA AAT CCT TGG-3') and an annealing temperature of 57˚C. These primers reside within the putative deletion and therefore cannot distinguish between animals lacking the putative deletion and animals heterozygous for it. Presence of a PCR product in this assay indicates that *TYRP1* coding regions 6 and 7 are present on at least one chromosome. Absence of a PCR product in this assay indicates that *TYRP1* coding regions 6 and 7 are absent on both chromosomes. We were unable to design a PCR assay to detect the putative deletion in heterozygotes because the putative deletion extends into a discontinuity in the Burmese python reference genome. Presence or absence of a PCR product in this assay is provided in S6 Table.

### Protein sequence alignment

Protein sequences were aligned using Clustal Omega [113], using default parameters.

## Supporting information

**S1 Fig. Raw gel images.** (A) Raw image of the gel displayed in Fig 2. (B) Raw image of the gel displayed in Fig 3. (C) Raw image of the gel displayed in Fig 4. Brightness and contrast settings

have not been adjusted in these images. x, experiment unrelated to the current study.
(TIF)

**S1 Table. Variants used in the Albino association study and primers to genotype these variants.**
(DOCX)

**S2 Table. Conservation of gene structure of melanogenesis genes.**
(DOCX)

**S3 Table. Primers to amplify and sequence coding regions of melanogenesis genes.**
(DOCX)

**S4 Table. Primers to amplify non-coding regions of *TYR*.**
(DOCX)

**S5 Table. Primers to investigate the putative deletion in *TYRP1*.**
(DOCX)

**S6 Table. Genotypes.**
(DOCX)

## Acknowledgments

We thank Matt Rockman and Katy Greenwald for advice on haplotype reconstruction; Bob Winning, Anne Casper, David Kass, and two anonymous reviewers for comments on the manuscript; and the Educational Course Support program of New England BioLabs for reagents used in undergraduate teaching labs. We thank the following individuals for contributing ball python sheds: Adam and Nicole Schmid; Alycia Butler; Amanda Hall; Andelyn Czajka; Brad Carter of Driftless Reptiles; Bryan Rivera; Chiron Graves; Chun Ku of Dynasty Reptiles; Dale Porcher; Daniel Ross; David Wolf of Tornado Alley Reptiles; David Burstein; Dawn and Kelsi Greene of Super Natural Balls; Dayna Plehn; Debby Brauer; Epic Vibrant Balls; Eric Chung of Chung Reptiles; Erin Burt; George Straub; Haily McCullough; Jaden Christensen; Jake Lewis; Jamie Palazzo of New Day Reptiles; Jeff Kearns; Jeff Linton; Jodi Wilkowski; Joe Myers; John Cordone of Blue Water Reptiles; Jordan Noland; Justin Kobylka of J. Kobylka Reptiles; Lindsay VanOrman; Lisa Huis; Manuel San Juan; Mark Bilger; Lynnet Melton; Maryann Barbon; Mia Hynes; Michael, Lisa, and Bodie Cole of Ballroom Pythons South; Morgan Evans and Michael Kitto of MK Pythons; Morgan Shelton; Paul and Amber Fiorito of Vivid Scales; Pets 'n' Things of Saline, MI; Rachel Voyt; Royal Black Balls; Ryan Boyd and Brittney Delacruz; Ryan Young of Molecular Reptile; Sergio McDole; Stephanie S. Crisp; Steve P. of Prime Pets; Zac Parpart; and several anonymous contributors. We thank the following individuals for providing images of color morphs: Beth Woodyard; Cat Church; Cormier Jason; Christine Miller; Darin Taylor; Daniel Hatcher; David C. Callahan; Donald Grinstead; Elijah Snyder; Innovative Ectotherms; Jake Lewis; Jessica Allison; Jessica Van Riper; Justin Kobylka; James Thompson; Justin Revington; Mariette van Vuuren; Mark Hopkins; Mark Smith; Matthew Lopez; Michael Freedman of The Florida Reptile Ranch; Morgan Evans; Phil Barclay, Robert Cooper; Ron Heisenberg; Ryan Young; Selectively Bred Serpents; Seb Des Légendes Celtiques; and Thananan Jivaramonaikul.

Most of the data in this study were collected by undergraduates enrolled in a laboratory course at Eastern Michigan University. These students constitute the BIO306W Consortium. These students were Alexandra Ernst, Alia Frederick, Alissa Zoltowski, Amber Northcutt, Andraya Ackerman, Anna Pathammavong, Annette Miller, Ashly Matzek, Asra Akhlaq,

Aubrey Martin, Bailey Knight, Benjamin Huff, Beth Wasserman, Brian Donald Condron, Caleb Sommer, Cassandra Rigor, Charles Southwell, Chase Chitwood, Chelsea Brown, Christina Roka, Ciarra Womack, Clay McKenzie, Daniela Nappo, Darby Fracassa, Deirdre McCarter, Dhruvalkumar Patel, Dominic Paoletti, Drake Dzierwa, Erica R. Geml, Erin Bissett, Ezekiel Butcher, Garrett Chance, Garry Lewis, Genesis Garmendia, Geo Pullockaran, Hajer Musa Abuzir, Haley Praski, Hanan Alroaini, Iqra Akhlaq, Ismael Yasin, Janelle Aethyr, Janelle Janisse, Jayce Alee Perysian, Jemar Rooks, Jonathan Chang, Jonathan Harris, Joseph H. Oberlitner, Joshua Mason, Juwan Taylor, Kailynn Sparks, Karissa Urban, Karli Siefman, Kealy Szymanski, Kelsy Roque, Keyan Marshall, Khaled Ali, Karleigh Hassenzahl, Kylie Powrie, Lauren Colone, Lissette Rosas, Manoj Perumallapalli, Mariam Samir, Maryam Nimer, Maya Mackey, Megan McNulty, Mel Roberts, Micaela Schempf, Molly Cook, Myah Kelly, Nahiel Sukar, Natalie Diaz, Natasha MacKay, Nathan Barnett, Nathaniel Gonzalez, Noura Taybeh, Pablo De la Vega, Rida Ali, Ronnie Bryans, Ryan Elliott, Saja Hussein, Samantha Glowacki, Samuel Teener, Sarah Holtzen, Sarah Schmidt, Shanti Bernstein, Shelan Mizuree, Smarpita Singh, Stevie Zabrosky, Taia Broadbent, Tommiea Robertson, Tyler Schallhorn, Verginio Persicone, William Soder, Wolfgang Ebersole, and Yvette Campbell.

## Author Contributions

**Conceptualization:** Chiron W. Graves, Hannah S. Seidel.

**Data curation:** Hannah S. Seidel.

**Formal analysis:** Autumn R. Brown, Hannah S. Seidel.

**Funding acquisition:** Autumn R. Brown, Hannah S. Seidel.

**Investigation:** Autumn R. Brown, Kaylee Comai, Dominic Mannino, Haily McCullough, Yamini Donekal, Hunter C. Meyers, Hannah S. Seidel.

**Methodology:** Hannah S. Seidel.

**Project administration:** Hannah S. Seidel.

**Resources:** Chiron W. Graves, Hannah S. Seidel.

**Supervision:** Hannah S. Seidel.

**Writing – original draft:** Hannah S. Seidel.

**Writing – review & editing:** Autumn R. Brown, Chiron W. Graves, Hannah S. Seidel.

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
