## [Decision Letter · Decision Letter 0]

17 Sep 2021

PONE-D-21-16582A community-science approach identifies genetic variants associated with three color morphs in ball pythons (*Python regius*)PLOS ONE

Dear Dr. Seidel,

Thank you for submitting your manuscript to PLOS ONE. After careful consideration, we feel that it has merit but does not fully meet PLOS ONE’s publication criteria as it currently stands. Therefore, we invite you to submit a revised version of the manuscript that addresses the points raised during the review process. The reviewers have indicated phrasing changes and clarification that is required prior to publication.

We look forward to receiving your revised manuscript.

Kind regards,

Brian W Davis

Academic Editor

PLOS ONE

Journal Requirements:

“This work was supported by a Faculty Research Fellowship and James H. Brickley Award from Eastern Michigan University to HSS and an Undergraduate Research Stimulus Program Award and a Don Brown and Meta Hellwig Undergraduate Research Award from Eastern Michigan University to ARB. The funders had no role in study design, data collection and analysis, decision to publish, or preparation of the manuscript”

“This work was supported by a Faculty Research Fellowship and James H. Brickley Award from Eastern Michigan University to HSS and an Undergraduate Research Stimulus Program Award and a Don Brown and Meta Hellwig Undergraduate Research Award from Eastern Michigan University to ARB. The funders had no role in study design, data collection and analysis, decision to publish, or preparation of the manuscript.”

5. PLOS ONE now requires that authors provide the original uncropped and unadjusted images underlying all blot or gel results reported in a submission’s figures or Supporting Information files. This policy and the journal’s other requirements for blot/gel reporting and figure preparation are described in detail at https://journals.plos.org/plosone/s/figures#loc-blot-and-gel-reporting-requirements and https://journals.plos.org/plosone/s/figures#loc-preparing-figures-from-image-files. When you submit your revised manuscript, please ensure that your figures adhere fully to these guidelines and provide the original underlying images for all blot or gel data reported in your submission. See the following link for instructions on providing the original image data: https://journals.plos.org/plosone/s/figures#loc-original-images-for-blots-and-gels. In your cover letter, please note whether your blot/gel image data are in Supporting Information or posted at a public data repository, provide the repository URL if relevant, and provide specific details as to which raw blot/gel images, if any, are not available. Email us at plosone@plos.org if you have any questions.

6. Please include a separate caption for each figure in your manuscript.

Reviewers' comments:

Reviewer's Responses to Questions

**Comments to the Author**

1. Is the manuscript technically sound, and do the data support the conclusions?

Reviewer #1: Yes

Reviewer #2: Yes

2. Has the statistical analysis been performed appropriately and rigorously? 

Reviewer #1: Yes

Reviewer #2: Yes

3. Have the authors made all data underlying the findings in their manuscript fully available?

Reviewer #1: Yes

Reviewer #2: Yes

4. Is the manuscript presented in an intelligible fashion and written in standard English?

Reviewer #1: Yes

Reviewer #2: Yes

5. Review Comments to the Author

Reviewer #1: Brown et al. presents a thorough investigation into the genetics behind several ball python color morphologies (Albino, Lavender Albino, and Ultramel). Utilizing a community-science based, non-invasive approach to collecting DNA samples from shed skin, they were able to find putative loss-of-function mutations for each of the described color morphs. Provided their analysis was based on the highly fragmented, scaffold level assembly of the Burmese python, the authors did an nice job validating each of the gene annotations with other existing sources. Their analysis of each color morph was detailed, organized, and thoroughly explained results even when a clear answer was not found. This was evident by the multiple approaches used to attempt to identify a putative causal variant for the TYR-albino haplotype. Although they were unable to identify a specific causal variant, their analysis was thorough and convincing to show there was no clear answer within the scope of this study.

In all, I feel that Brown et al. made convincing arguments for all color morphologies described and their claims are supported by empirical data, statistical analyses, and previous literature. Given a portion of the study was conducted by undergraduates, they implemented stringent controls and checks throughout the analysis to ensure the data was up to standard and could be confidently included into the study. Their discussion and methods are well written and easy to follow, but the discussion may benefit from a few comments on how the fragmented Burmese python reference may have impacted their ability to accurately assembly genes fragmented across several scaffolds and the subsequent identification of causal variants from those fragmented genes. Especially with the identification of a variable number tandem repeat within a discontinuous region of the Burmese python assembly. Beyond the few comments below, Brown et al. make compelling arguments for the first analysis of multiple color morphologies in ball pythons.

1. On page 8, line 142 it says "These results demonstrate and association between…" I'm assuming they meant "demonstrated an association"

2. On page 9, line 148 it says " We found that one of these animala was homozygous…" I'm assuming they meant "animals".

3. On page 13, line 254 it states "We found that all four Lavender Albinos were homozygous for the deletion (Figure 3C-D)". However, Figure 3D shows there are 5 homozygous Lavender Albinos and the legend also states there are 5. Please clarify how many Lavender Albinos were used.

Reviewer #2: This study investigates the genetic underpinnings of three color mutations in a common pet snake, the ball python, Python regius. Using known loss-of-function mutations across a variety of vertebrate species, the authors identified a suite of genes from which the Burmese python genome could be used for primer design. Three mutations were identified corresponding to the albino mutation, one for the lavender albino variant, and both a mutation and a deletion were found in the ultramel variant. The study is novel, and the student center approach is commendable. My concerns are minor.

1) The sample sizes for the color variants are very low. E.g., 20 albino, 5 albino/candy, 1 albino/toffee, and 1 toffee; 5 lavender albinos, 3 ultramels. As such. The confidence in identifying the causal mutations may also be low.

2) The definition of unrelated is relatively weak. Obtaining samples from different breeders does not make the animals unrelated. In fact, many of the breeders listed as assisting commonly source animals from one or just a few other breeders. Therefore, there is likely a lot of shared recent ancestry. The authors should therefore consider generating a more thorough pedigree for each sample. Basically, to which origin can the variants be traced back to.

3) I feel that the sample size for the ultramel variant is incredibly low. Given both the convoluted history of that variant (at least three imported animals over the space of 10 to 15 years), and uncertainty that these animals labelled as “ultramel” are compatible (i.e., will produce an “ultramel” variant if crossed), means that the result is a little speculative. I suppose it could be framed as “animals labelled by a breeder as ultramel exhibited two distinct alleles”, not “the ultramel trait is underpinned by two alleles”. The reason I say this is that one of these alleles might correspond to one of the previously important animals with a similar phenotype, and over time, that has been lumped in as being an “ultramel”. The authors should therefore maybe consider either 1) obtaining more samples of “ultramel” animals, or 2) obtaining samples from the other lines with a similar trait (e.g., the monarch trait and Crider trait). It might simply be case of mislabelling (known or unknown) by the breeder.

4) I am disappointed that the authors did not include animals heterozygous for these traits, which are all readily available in collections. This would help confirm the diagnostic nature of the study.

That said, my concerns are relatively minor and the study identifies genes and putative alleles for these three traits.

Minor comments:

Ln 158 – The different albino variants needs to be identified here. I recommend using pictures. Toffee and Candy variants are phenotypically identical but are from different origins. I agree with the conclusion that they are the same variant. This is likely the case with many color and pattern variants in this species.

Ln 383 – Caramel albinos are not necessarily infertile, although a reduced fertility rate has been observed. The main issues was that with some lines of caramel albino (they have been imported several times form the wild), the offspring exhibited severe spinal kinking, and hence non-viable.

6. PLOS authors have the option to publish the peer review history of their article (what does this mean?). If published, this will include your full peer review and any attached files.

Reviewer #1: No

Reviewer #2: No

---

## [Author Response · Author response to Decision Letter 0]

28 Dec 2021

Responses to Reviewers

We thank the reviewers for their helpful comments in the manuscript. Their comments have improved the manuscript and strengthened the conclusions of our study.

Comments to the Author

Reviewer #1: Brown et al. presents a thorough investigation into the genetics behind several ball python color morphologies (Albino, Lavender Albino, and Ultramel). Utilizing a community-science based, non-invasive approach to collecting DNA samples from shed skin, they were able to find putative loss-of-function mutations for each of the described color morphs. Provided their analysis was based on the highly fragmented, scaffold level assembly of the Burmese python, the authors did an nice job validating each of the gene annotations with other existing sources. Their analysis of each color morph was detailed, organized, and thoroughly explained results even when a clear answer was not found. This was evident by the multiple approaches used to attempt to identify a putative causal variant for the TYR-albino haplotype. Although they were unable to identify a specific causal variant, their analysis was thorough and convincing to show there was no clear answer within the scope of this study.

In all, I feel that Brown et al. made convincing arguments for all color morphologies described and their claims are supported by empirical data, statistical analyses, and previous literature. Given a portion of the study was conducted by undergraduates, they implemented stringent controls and checks throughout the analysis to ensure the data was up to standard and could be confidently included into the study. Their discussion and methods are well written and easy to follow, but the discussion may benefit from a few comments on how the fragmented Burmese python reference may have impacted their ability to accurately assembly genes fragmented across several scaffolds and the subsequent identification of causal variants from those fragmented genes. Especially with the identification of a variable number tandem repeat within a discontinuous region of the Burmese python assembly. Beyond the few comments below, Brown et al. make compelling arguments for the first analysis of multiple color morphologies in ball pythons.

Thanks for this suggestion. We have added a paragraph to the Discussion section describing the impact of the fragmented Burmese python reference genome on our study. Main impacts of the fragmented Burmese python reference genome included (i) inability to interrogate TYR intron 1 in the TYRAlbino allele, and (ii) inability to the map the breakpoints of the putative deletion in TYRP1. The variable number tandem repeat was not affected by the fragmentation of the reference genome because this repeat resides in a region of the reference genome that was not fragmented. Our new paragraph is provided here:

"One challenge for genetic studies in ball python is the absence of a reference genome. To fill this gap, we relied on the genome of Burmese python [90], which is a scaffold-level assembly. Contigs in this assembly are small, and genes are sometimes fragmented across scaffolds. This fragmentation limited our analysis of the TYRAlbino haplotype and the putative deletion in TYRP1. For TYRAlbino, we were unable to assess whether the TYRAlbino haplotype contained a large indel or rearrangement in the first intron of TYR, due to a discontinuity in the Burmese python genome in this region. For the putative deletion in TYRP1, we were unable to map the exact breakpoints of the deletion, due to a discontinuity in the Burmese python genome immediately downstream of TYRP1. As genomic resources for Burmese python and ball python expand, we expect to identify a putatively causative variant on the TYRAlbino haplotype and to map the breakpoints of the putative deletion in TYRP1. This information will increase the prospects for genetic testing for these alleles."

In response to the concern about gene annotations, we have updated the Methods section to more clearly describe the conservation of gene structure across species. Gene structure for the genes analyzed in this study were highly conserved. Genes contained the same number of exons in all species, and the intron-exon boundaries were perfectly conserved across species (with the exception of one slight difference of one boundary for one gene in mouse). Moreover, every coding region of every gene was present in the Burmese python reference assembly. It is true that some genes were split across scaffolds, but these splits always occurred in introns. Thus, while it remains theoretically possible that our new gene annotations contain minor errors, our confidence in these gene annotations is high. Our new passage is provided here:

"A main cause of these errors was that the Burmese python genome (Python_molurus_bivittatus-5.0.2) is a scaffold-level assembly; many genes were split across scaffolds. We therefore curated new gene annotations in Burmese python, using conservation of gene structure across species. Alignment of protein sequences for each gene from corn snake, anole lizard, chicken, and/or mouse revealed that gene structure was highly conserved for all seven genes: Genes in each species contained the same number of coding regions, and the coding-region boundaries relative to protein sequences were perfectly conserved across species, with the exception of one slight difference in mouse for one boundary of SLC45A2 (S2 Table). We therefore felt confident that conservation of coding-region boundaries could be used to curate new gene annotations in Burmese python. While it remains theoretically possible that our new gene annotations contain minor errors, our confidence in these gene annotations is high, given the high conservation of gene structure across species."

On page 8, line 142 it says "These results demonstrate and association between…" I'm assuming they meant "demonstrated an association".

Thank you for finding this typo. We have corrected it.

On page 9, line 148 it says " We found that one of these animala was homozygous…" I'm assuming they meant "animals".

Thank you for finding this typo. We have corrected it.

On page 13, line 254 it states "We found that all four Lavender Albinos were homozygous for the deletion (Figure 3C-D)". However, Figure 3D shows there are 5 homozygous Lavender Albinos and the legend also states there are 5. Please clarify how many Lavender Albinos were used.

This table includes the original Lavender Albino animal in which the OCA2 deletion was identified. We have updated the figure caption to indicate the inclusion of this animal. The same is true for the Ultramel table in Figure 4. We have updated the Figure 4 caption in the same way.

Reviewer #2: This study investigates the genetic underpinnings of three color mutations in a common pet snake, the ball python, Python regius. Using known loss-of-function mutations across a variety of vertebrate species, the authors identified a suite of genes from which the Burmese python genome could be used for primer design. Three mutations were identified corresponding to the albino mutation, one for the lavender albino variant, and both a mutation and a deletion were found in the ultramel variant. The study is novel, and the student center approach is commendable. My concerns are minor.

1) The sample sizes for the color variants are very low. E.g., 20 albino, 5 albino/candy, 1 albino/toffee, and 1 toffee; 5 lavender albinos, 3 ultramels. As such. The confidence in identifying the causal mutations may also be low.

We have increased the sample size for each morph. The Albino/Toffee/Candy sample was increased from 27 to 50. The Lavender Albino sample was increased from 5 to 14. The Ultramel sample was increased from 3 to 11. These expanded sample sizes support the conclusions in the original manuscript. 

We recognize that larger samples would be preferable. However, obtaining additional samples was not possible. We have collected over 1,400 samples from owners and breeders across the US. Our small sample sizes reflect the rareness of the Albino/Toffee/Candy, Lavender Albino, and Ultramel morphs. 

We cannot exclude the possibility that additional alleles exist for the morphs described in this manuscript. However, such alleles are necessarily rare, given their absence in our sample. We have amended out Discussion section to include the following statement, "Due to the small sample size of our study, we cannot exclude the possibility of additional loss-of-function alleles of these genes. However, such alleles are expected to be at low frequency in the ball python population, given their absence from our sample."

2) The definition of unrelated is relatively weak. Obtaining samples from different breeders does not make the animals unrelated. In fact, many of the breeders listed as assisting commonly source animals from one or just a few other breeders. Therefore, there is likely a lot of shared recent ancestry. The authors should therefore consider generating a more thorough pedigree for each sample. Basically, to which origin can the variants be traced back to.

This argument is a fair point. We have removed the word ‘unrelated’ from the manuscript. We have added a description to the Methods Section justifying our decision to limit our sample to animals contributed by different contributors. This description indicates that our sampling strategy was designed to increase genetic diversity within each morph. Our description also indicates that although we cannot exclude that some animals in our sample might be close relatives, our discovery of multiple alleles for two morphs (Albino and Ultramel) indicates that our samples include animals from different lineages. In addition, our description indicates that generating a thorough pedigree for each animal was not possible because most owners lacked this information. Our revised passage is provided below:

“To maximize genetic diversity within each category of morph, we limited our sample of animals of any one morph to animals contributed by different contributors. Exceptions were made if animals contributed by the same contributor had been obtained from different breeders. The goal of this strategy was to reduce the number of animals that were close relatives (e.g. siblings or parent-offspring pairs). Obtaining full pedigree information was not possible because most owners lacked this information (e.g. “I bought my animal at a pet store” or “I got my animal from a friend who was moving away”). Our discovery of multiple alleles for the Albino and Ultramel color morphs supports the idea that animals in our sample were derived from multiple lineages. However, we cannot exclude the possibility that some animals in our sample may have been close relatives.”

3) I feel that the sample size for the ultramel variant is incredibly low. Given both the convoluted history of that variant (at least three imported animals over the space of 10 to 15 years), and uncertainty that these animals labelled as “ultramel” are compatible (i.e., will produce an “ultramel” variant if crossed), means that the result is a little speculative. I suppose it could be framed as “animals labelled by a breeder as ultramel exhibited two distinct alleles”, not “the ultramel trait is underpinned by two alleles”. The reason I say this is that one of these alleles might correspond to one of the previously important animals with a similar phenotype, and over time, that has been lumped in as being an “ultramel”. The authors should therefore maybe consider either 1) obtaining more samples of “ultramel” animals, or 2) obtaining samples from the other lines with a similar trait (e.g., the monarch trait and Crider trait). It might simply be case of mislabelling (known or unknown) by the breeder.

We have increased our sample size of Ultramels from 3 to 11. These 11 animals included six animals homozygous for the TYRP1 R305H missense variant, three animals homozygous for the TYRP1 putative deletion, and two compound heterozygotes. Thus, this larger sample size supports the conclusion that animals described as Ultramel are a mixture of multiple genotypes, but are always homozygous or compound heterozygous for loss-of-function alleles of TYRP1.

A major difficulty in studying the genetics of ball pythons is that the origins of ball python morphs are not well documented in the scientific literature. We have heard multiple stories about the origins of Ultramel morph, but none of these stories have been officially documented. Likewise, we have heard multiple thoughts about whether Ultramel is allelic with morphs having similar coloration (e.g. Monarch, Crider). We have attempted to obtain samples from these other morphs, but breeders have been unwilling to send samples. (In some cases, breeders have shared with us that they are purposefully withholding samples out of concern that our discoveries might decrease the value of their animals). Thus, it remains possible that one or more of these morphs is allelic with Ultramel. This possibility does not conflict with our conclusions that the Ultramel morph is caused by multiple loss-of-function alleles of TYRP1.

4) I am disappointed that the authors did not include animals heterozygous for these traits, which are all readily available in collections. This would help confirm the diagnostic nature of the study.

We purposefully excluded heterozygotes from our study because we have found that animals described as heterozygous for recessive morphs are not always heterozygous for these morphs. This situation likely arises from mislabeling by owners and breeders (intentional or non-intentional), perhaps in some cases from mis-attribution of paternity. We have added passages to the Methods Section justifying our exclusion of animals described as heterozygotes. The relevant passage is below:

“The total set of animals comprised 50 Albinos, 14 Lavender Albinos, 11 Ultramels, one animal described as heterozygous for Lavender Albino, and 46 animals described as having normal coloration (i.e. wildtype) or belonging to color morphs other than Albino, Lavender Albino, or Ultramel. Phenotypes of Albinos, Lavender Albinos, and Ultramels were confirmed by examining shed skins for a reduction of brown-to-black coloration. The single animal described as heterozygous for Lavender Albino was included to demonstrate the three-primer assay for genotyping the OCA2 deletion. Heterozygotes for Albino or Ultramel and additional heterozygous for Lavender Albino were excluded from our study because we have found that animals described as heterozygous for recessive traits by breeders and owners are sometimes not heterozygous for these traits (e.g. due to mis-attribution of paternity).”

That said, my concerns are relatively minor and the study identifies genes and putative alleles for these three traits.

Thanks for the positive feedback.

Minor comments:

Ln 158 – The different albino variants needs to be identified here. I recommend using pictures. Toffee and Candy variants are phenotypically identical but are from different origins. I agree with the conclusion that they are the same variant. This is likely the case with many color and pattern variants in this species.

Thanks for this suggestion. We have updated Figures 2 and 5 to include images of Candy and Toffee animals.

Ln 383 – Caramel albinos are not necessarily infertile, although a reduced fertility rate has been observed. The main issues was that with some lines of caramel albino (they have been imported several times form the wild), the offspring exhibited severe spinal kinking, and hence non-viable.

Thanks for the additional information about Caramel Albinos. We have updated the passage about Caramel Albinos to indicate that multiple lines of this morph have been identified and that defects have included spinal kinking and reduced female fertility. Our updated passage is included here:

“Our finding of multiple TYRP1 alleles for the Ultramel color morph was unexpected. This morph was not previously described as having multiple alleles. We speculate that one of the two alleles associated with the Ultramel color morph may represent an allele originally associated with a morph known as Caramel Albino. Multiple lineages of Caramel Albinos have been described, and their coloration similar to Ultramels. Caramel Albinos have been disfavored among owners and breeders because of spinal kinking and reduced female fertility. Caramel Albinos were not included in our study because we were unable to recruit any Caramel Albino samples. We speculate that the Caramel Albino morph may be allelic with Ultramel, and that one of two TYRP1 alleles associated with Ultramel may represent an allele originally described as Caramel Albino. Alternately, Ultramel may be distinct from Caramel Albino, and the two TYRP1 alleles may represent two distinct origins of the Ultramel morph.”

---

## [Decision Letter · Decision Letter 1]

26 Apr 2022

PONE-D-21-16582R1A community-science approach identifies genetic variants associated with three color morphs in ball pythons (*Python regius*)PLOS ONE

Dear Dr. Seidel,

Thank you for submitting your manuscript to PLOS ONE. After careful consideration, we feel that it has merit but does not fully meet PLOS ONE’s publication criteria as it currently stands. Therefore, we invite you to submit a revised version of the manuscript that addresses the points raised during the review process.

We look forward to receiving your revised manuscript.

Kind regards,

Brian W Davis

Academic Editor

PLOS ONE

Journal Requirements:

Reviewers' comments:

Reviewer's Responses to Questions

**Comments to the Author**

1. If the authors have adequately addressed your comments raised in a previous round of review and you feel that this manuscript is now acceptable for publication, you may indicate that here to bypass the “Comments to the Author” section, enter your conflict of interest statement in the “Confidential to Editor” section, and submit your "Accept" recommendation.

Reviewer #1: (No Response)

2. Is the manuscript technically sound, and do the data support the conclusions?

Reviewer #1: Yes

3. Has the statistical analysis been performed appropriately and rigorously? 

Reviewer #1: Yes

4. Have the authors made all data underlying the findings in their manuscript fully available?

Reviewer #1: No

5. Is the manuscript presented in an intelligible fashion and written in standard English?

Reviewer #1: Yes

6. Review Comments to the Author

Reviewer #1: The authors have provided a revised version of their manuscript, answering all of my comments as well as the other reviewers comments in a manner that in my opinion satisfies the changes required for acceptance. The only minor revision that remains is the availability of their sequence data on GenBank. The authors provide GenBank accession numbers (MZ269492-MZ269502), but the sequences do not appear to be publicly available yet as no results are returned when queried. Once the data is fully available and accounted for, I feel the manuscript is ready to proceed to publication.

7. PLOS authors have the option to publish the peer review history of their article (what does this mean?). If published, this will include your full peer review and any attached files.

Reviewer #1: No

---

## [Author Response · Author response to Decision Letter 1]

16 May 2022

Thanks for noticing that GenBank had failed to release the data associated with our publication. We have contacted GenBank and fixed the problem. The sequences associated with accession numbers MZ269492-MZ269502 are now publicly available. Thus, our publication now meets the data availability policy of PLoS ONE.

---

## [Decision Letter · Decision Letter 2]

26 Sep 2022

PONE-D-21-16582R2A community-science approach identifies genetic variants associated with three color morphs in ball pythons (*Python regius*)PLOS ONE

Dear Dr. Seidel,

Thank you for submitting your manuscript to PLOS ONE. After careful consideration, we feel that it has merit but does not fully meet PLOS ONE’s publication criteria as it currently stands. Therefore, we invite you to submit a revised version of the manuscript that addresses the points raised during the review process. Please submit your revised manuscript by Nov 10 2022 11:59PM. If you will need more time than this to complete your revisions, please reply to this message or contact the journal office at plosone@plos.org. Please include the following items when submitting your revised manuscript:A rebuttal letter that responds to each point raised by the academic editor and reviewer(s). You should upload this letter as a separate file labeled 'Response to Reviewers'.A marked-up copy of your manuscript that highlights changes made to the original version. You should upload this as a separate file labeled 'Revised Manuscript with Track Changes'.An unmarked version of your revised paper without tracked changes. You should upload this as a separate file labeled 'Manuscript'.If applicable, we recommend that you deposit your laboratory protocols in protocols.io to enhance the reproducibility of your results. Protocols.io assigns your protocol its own identifier (DOI) so that it can be cited independently in the future. For instructions see: https://journals.plos.org/plosone/s/submission-guidelines#loc-laboratory-protocols. Additionally, PLOS ONE offers an option for publishing peer-reviewed Lab Protocol articles, which describe protocols hosted on protocols.io. Read more information on sharing protocols at https://plos.org/protocols?utm_medium=editorial-email&utm_source=authorletters&utm_campaign=protocols.

We look forward to receiving your revised manuscript.

Kind regards,

Xiaolin Bi, Ph.D.

Academic Editor

PLOS ONE

Journal Requirements:

Reviewers' comments:

Reviewer's Responses to Questions

**Comments to the Author**

1. If the authors have adequately addressed your comments raised in a previous round of review and you feel that this manuscript is now acceptable for publication, you may indicate that here to bypass the “Comments to the Author” section, enter your conflict of interest statement in the “Confidential to Editor” section, and submit your "Accept" recommendation.

Reviewer #1: All comments have been addressed

Reviewer #3: (No Response)

2. Is the manuscript technically sound, and do the data support the conclusions?

Reviewer #1: (No Response)

Reviewer #3: Yes

3. Has the statistical analysis been performed appropriately and rigorously? 

Reviewer #1: (No Response)

Reviewer #3: N/A

4. Have the authors made all data underlying the findings in their manuscript fully available?

Reviewer #1: (No Response)

Reviewer #3: Yes

5. Is the manuscript presented in an intelligible fashion and written in standard English?

Reviewer #1: (No Response)

Reviewer #3: Yes

6. Review Comments to the Author

Reviewer #1: (No Response)

Reviewer #3: The authors describe putative variants in three genes associated with different color morphs of the ball python. The study is well described, methodology detailed, and overall interesting for the field. It shows another example of a succesful collaboration between public and researchers using the citizen science approach. I also find it very cool that the authors include and acknowledge the undergraduate students contribution. However, I found a few ambiguities and formal genetic issues that should be corrected before publication.

Fig. 1 is the first place talking about "Candy" and "Toffee". These should be mentioned and explained in the introduction (around L86/87). Are they variations of "Albino"? From looking at this Figure, they seem to be visually closer to the "Ultramel" as they both seem to have darker and more brownish coloration between the yellow spots, compared to the white "Albino" or kind of cream in "Lavender Albino". Why are they considered albino? This needs to be explained in the introduction section. -edit: I have found the relevant section in L218-224. That is way too late, please move that paragraph into Introduction.

Table 1 lacks some identified species, e.g. multiple TYRP1 variants in dog are well known (https://www.omia.org/OMIA001249/). Also, the dog reference (78) is incorrectly assigned to SLC24A5, instead of SLC45A2, where it is missing (https://www.omia.org/OMIA001821/9615/). I suggest the authors use the the recent comprehensive review for the canine variants (10.1111/age.13154). It would be also desirable to link the table gene entries to OMIA database for full reference.

L116 "Similar phenotypes occur in other animals, where the loss of melanin extends to feathers and scales" should also include fur.

L139 What is "Non-Albino"? In the Methods section, these are not termed. I presume those are the "...normal coloration (i.e. wildtype) or belonging to color morphs...", so just define the term in Methods (L489).

Please clarify if there are 46 or 56 of these "Non-Albino" individuals. Fig. 2 and the paragraph above states 56 as opposed to 46 stated in the Methods L488 and labeled "Non" in the first column of Table S6. If you used other colors as controls, that needs to be clearly specified.

Fig. 2C should have title changed from "TYR genotype" to "TYR haplotype".

L152 Include some numbers here and refer to Fig. 2C. How many homozygotes and comp. heterozygotes? It should be clearly stated (without the need of opening supplementary file) that the non-albino animals were not found homozygous for the albino-associated haplotypes.

L167 (and throughout for all others) Define the variants using the correct nomenclature standard (https://varnomen.hgvs.org/recommendations/protein/variant/substitution/), at least for the first time mentioned. E.g. the D394G should read UPQ41330.1:p.(Asp394Gly). Additionally, this describes a protein variant; therefore, it cannot be "in the coding region of a gene" - please, rephrase the sentences accordingly. As an example, this sentence should read: "We found that one of these animals was homozygous for a missense variant in the third exon of TYR (MZ269492:c.A3695G), which leads to an aspartic acid-to-glycine exchange (UPQ41330.1:p.(Asp394Gly); termed hereafter D394G)."

L239-248 The first paragraph is not about results. I suggest moving the part between L242 and L246 into Discussion.

L281-293 The first paragraph is not about results. I suggest moving the part between L284 and L291 into Discussion.

L297 Same comment as above re nomenclature. The coding variant should be designated with bp position and the actual substitution. The amino acid exchange is the consequence of the exonic variant.

Fig. 4 The part of legend in (B) which states "Ultramels heterozygous for R305H...when heterozygous." should be moved in to the main Results text around L333. And it should be more explained. Were all the amplicons detected in the 2 heterozygous animals? Can you show the PCR picture of the heterozygote (in Fig. S1 would be enough)?

L337-339 remove the parentheses

L386 "slicing" should read splicing

L447 Current state-of-the-art methods (such as targeted high-throughput sequencing, whole-genome sequencing, or long-read technologies) should be mentioned here as an obvious solution to missing reference and/or structural variants. See (cite) e.g. https://www.mdpi.com/2073-4425/10/5/386/htm for an example of a coat color gene variant identification without correct reference sequence.

L454 "strip" should be stripes

L485 "supports" at the end of sentence should probably be deleted

L487 How many were "Toffee" or "Candy"?

L525 Please, specify the software used. Does TBLASTN refer to the NIH Translated BLAST tool? Citation should be provided, as well as clarification, if default parameters were used for the search or if anything was adjusted.

7. PLOS authors have the option to publish the peer review history of their article (what does this mean?). If published, this will include your full peer review and any attached files.

Reviewer #1: No

Reviewer #3: No

---

## [Author Response · Author response to Decision Letter 2]

2 Oct 2022

Responses to Reviewers (provided here and in attached document)

We thank the reviewer for their helpful comments on the manuscript. These comments have improved the clarity of writing and will help readers navigate our manuscript.

Reviewer #3: The authors describe putative variants in three genes associated with different color morphs of the ball python. The study is well described, methodology detailed, and overall interesting for the field. It shows another example of a succesful collaboration between public and researchers using the citizen science approach. I also find it very cool that the authors include and acknowledge the undergraduate students contribution. However, I found a few ambiguities and formal genetic issues that should be corrected before publication.

Fig. 1 is the first place talking about "Candy" and "Toffee". These should be mentioned and explained in the introduction (around L86/87). Are they variations of "Albino"? From looking at this Figure, they seem to be visually closer to the "Ultramel" as they both seem to have darker and more brownish coloration between the yellow spots, compared to the white "Albino" or kind of cream in "Lavender Albino". Why are they considered albino? This needs to be explained in the introduction section. -edit: I have found the relevant section in L218-224. That is way too late, please move that paragraph into Introduction.

Thank you for pointing out this lack of clarity regarding the phenotypic variability within the Albino color morph. We have addressed this issue in two ways. First, we have added a sentence to the caption of Figure 1 explaining that the Albino color morph is phenotypically variable. This sentence reads:

“Phenotypes within the Albino color morph are variable, with some animals having skin patches that are white and others having skin patches that are light beige.”

Second, we have modified the text introducing the Albino color morph to explain the phenotypic variability within the morph. This paragraph reads:

“The Albino color morph in ball pythons is characterized by an absence or near absence of melanin in the skin and eyes—the brown-to-black coloration observed in wildtype is absent or severely reduced, and skin patches appear white or light beige (Fig 1D-F). The Albino color morph is described by breeders as having three alleles (AlbAlbino, AlbCandy, and AlbToffee), although prior to the current study, it remained unclear whether these alleles represented distinct molecular variants of the same gene or the same molecular variant discovered independently three times. We began our analysis of the Albino color morph by treating all animals within the color morph as a single group.”

Table 1 lacks some identified species, e.g. multiple TYRP1 variants in dog are well known (https://www.omia.org/OMIA001249/). Also, the dog reference (78) is incorrectly assigned to SLC24A5, instead of SLC45A2, where it is missing (https://www.omia.org/OMIA001821/9615/). I suggest the authors use the the recent comprehensive review for the canine variants (10.1111/age.13154). It would be also desirable to link the table gene entries to OMIA database for full reference.

Thank you for helping us improve this table. We have added a reference to the suggested review article describing color phenotypes in dogs. We have also fixed the error for the reference of SLC45A2 in dogs. Lastly, we have added OMIA links for the five genes with OMIA pages.

L116 "Similar phenotypes occur in other animals, where the loss of melanin extends to feathers and scales" should also include fur.

We have revised the text accordingly. Thank you for the suggestion.

L139 What is "Non-Albino"? In the Methods section, these are not termed. I presume those are the "...normal coloration (i.e. wildtype) or belonging to color morphs...", so just define the term in Methods (L489).

We have added a text to the Methods section to define this term. This text reads, 

“We use the term Non-Albino to refer to animals having normal coloration or belonging to morphs other than Albino.”

Please clarify if there are 46 or 56 of these "Non-Albino" individuals. Fig. 2 and the paragraph above states 56 as opposed to 46 stated in the Methods L488 and labeled "Non" in the first column of Table S6. If you used other colors as controls, that needs to be clearly specified.

We have clarified this point as follows,

“The Non-Albinos included some Lavender Albinos and some Ultramels.”

Fig. 2C should have title changed from "TYR genotype" to "TYR haplotype".

We respectfully disagree. This portion of the figure shows diploid combinations of haplotypes, which constitute genotypes. Thus, the term genotype is the more appropriate term here.

L152 Include some numbers here and refer to Fig. 2C. How many homozygotes and comp. heterozygotes? It should be clearly stated (without the need of opening supplementary file) that the non-albino animals were not found homozygous for the albino-associated haplotypes.

Thank you for pointing out this lack of detail. We have revised the test accordingly. It now reads,

“The association for TYR was driven by two features of the haplotype distribution: (i) haplotype diversity was reduced from a total of 15 TYR haplotypes among Non-Albinos to three haplotypes among Albinos (Fig 2A, C), and (ii) two of the three TYR haplotypes found in Albinos were rare among Non-Albinos (Fig 2A). These results demonstrate an association between the Albino color morph and variants in TYR; further, they show that all Albino animals were homozygous (34 animals) or compound heterozygous (16 animals) for any of three haplotypes of TYR.”

Stating here that Non-Albinos were not found to be homozygous for the Albino-associated haplotypes is misleading because one of the Albino-associated haplotypes is actually quite common among Non-Albinos (Fig 2A). The reason is that this haplotype exists in two versions within the population. One version contains a missense variant that causes the Albino color morph (P384L). The other version does not contain this missense variant. At this point in the text, we have not discovered the missense variant and cannot distinguish between the two versions of the haplotype. Once we discover the missense variant (in the paragraphs immediately following the paragraph in question), we state that Non-Albinos were never found to be homozygous for the missense-containing version of the haplotype. We have revised this text to make it more it clear. The text reads, 

“We hypothesized that D394G, P384L, and an unidentified variant on the TYRAlbino haplotype were causative for the Albino color morph. This hypothesis predicted that all Albinos would be homozygous or compound heterozygous for these variants, whereas Non-Albinos would be heterozygotes and non-carriers. To test this prediction, we genotyped our full panel of 50 Albinos and 56 Non-Albinos for the missense variants D394G and P384L. Consistent with our prediction, we found that the Albinos were exclusively homozygous or compound heterozygous for D394G, P384L, or the TYRAlbino haplotype (Fig 2C). The Non-Albinos were exclusively heterozygotes or non-carriers (Fig 2C). ”

L167 (and throughout for all others) Define the variants using the correct nomenclature standard (https://varnomen.hgvs.org/recommendations/protein/variant/substitution/), at least for the first time mentioned. E.g. the D394G should read UPQ41330.1:p.(Asp394Gly). Additionally, this describes a protein variant; therefore, it cannot be "in the coding region of a gene" - please, rephrase the sentences accordingly. As an example, this sentence should read: "We found that one of these animals was homozygous for a missense variant in the third exon of TYR (MZ269492:c.A3695G), which leads to an aspartic acid-to-glycine exchange (UPQ41330.1:p.(Asp394Gly); termed hereafter D394G)."

Thank you for this suggestion. We have revised the text accordingly. It reads,

“We found that one of these animals was homozygous for a missense variant in the third coding region of TYR (MZ269492:c.A3695G), which leads to an aspartic acid-to-glycine exchange [UPQ41330.1:p.(Asp394Gly)]. This variant is termed hereafter D394G. A second animal was homozygous for a different missense variant, also in the third coding region of TYR (MZ269492:c.C3665T), which leads to a proline-to-leucine exchange [UPQ41330.1:p.(Pro384Gly)]. This variant is termed hereafter P384L. .“

L239-248 The first paragraph is not about results. I suggest moving the part between L242 and L246 into Discussion.

We respectively disagree that this paragraph should be moved to the Discussion. The purpose of this paragraph is to introduce the hypothesis that the Lavender Albino color morph is caused by loss of function of OCA2. We have rewritten this paragraph to make its purpose clearer,

“The Lavender Albino color morph is characterized by skin patches that are lavender instead of brown or black. This phenotype is thought to arise from melanin levels that are dramatically reduced but not entirely eliminated. We hypothesized that this phenotype was caused by loss of function of OCA2, which encodes a chloride channel required for maintaining the pH of melanosomes [88,95]. When the OCA2 protein is absent or non-functional, the enzymes that synthesize melanin are less active, and only small amounts of melanin are produced [96,97]. The resulting phenotype is typically intermediate in severity between the loss-of-function phenotypes of TYR versus other melanogenesis genes (Table 1). OCA2 was therefore a good candidate for the causative gene of the Lavender Albino color morph.”

L281-293 The first paragraph is not about results. I suggest moving the part between L284 and L291 into Discussion.

We respectively disagree that this paragraph should be moved to the Discussion. The purpose of this paragraph is to introduce the hypothesis that the Ultramel color morph is caused by loss of function of TYRP1, TYRP2, SLC7A11, SLC24A5, or SLC45A2. We have rewritten this paragraph to make its purpose clearer,

“The Ultramel color morph is characterized by skin patches that are tan or light brown, rather than dark brown or black. This phenotype suggests a mild loss of melanin. We hypothesized that this phenotype was caused by loss of function of one of five genes: TYRP1, TYRP2, SLC7A11, SLC24A5, and SLC45A2 (Table 1). TYRP1 and TYRP2 encode enzymes involved in synthesizing melanin [87]. Their loss-of-function phenotypes are mild because of partially redundancy with other enzymes in the melanin synthesis pathway. SLC7A11 encodes a transporter responsible for importing cystine into the cell [99]. Cystine is a precursor to some forms of melanin, and its reduction alters melanin levels [77]. SLC24A5 encodes a K+-dependent Na+-Ca2+ exchanger [89]. SLC45A2 encodes a putative sugar transporter [90]. Loss of their encoded proteins reduces melanin through mechanisms that may involve defects in regulation of melanosome pH [84,89,100]. These fives genes were therefore good candidates for the causative gene of the Ultramel color morph.”

L297 Same comment as above re nomenclature. The coding variant should be designated with bp position and the actual substitution. The amino acid exchange is the consequence of the exonic variant.

Thank you for this suggestion. We have revised the text accordingly. It reads,

“Comparison of these sequences to wildtype revealed a single coding variant: a missense variant in the fourth coding region of TYRP1 (MZ269497:c.G1720A), which leads to an arginine-to-histidine exchange [UPQ41334.1:p.Arg305His]. The Ultramel animal was homozygous for this variant, termed hereafter R305H.”

Fig. 4 The part of legend in (B) which states "Ultramels heterozygous for R305H...when heterozygous." should be moved in to the main Results text around L333. And it should be more explained. Were all the amplicons detected in the 2 heterozygous animals? Can you show the PCR picture of the heterozygote (in Fig. S1 would be enough)?

Thanks for encouraging us to add more detail related to the Ultramels heterozygous for R305H. We have revised the main text to explain why the deletion cannot be detected in heterozygotes. The text reads,

“Loss of these coding regions was specific to Ultramels lacking the R305H variant and did not occur in any of the Non-Ultramels, nor in the Ultramels homozygous or heterozygous for R305H (Fig 4B). A simple explanation of this pattern is that the Ultramels lacking R305H were homozygous for a deletion spanning coding regions 6 and 7 of TYRP1. The two Ultramels heterozygous for R305H were presumed to be compound heterozygous for this putative deletion, although we were not able to confirm this heterozygosity because the putative deletion extended into a discontinuity in the Burmese python reference genome (and therefore could not be detected in heterozygotes).” 

The genotypes of the two heterozygous animals are provided in S6 Table, which indicates presence/absence of a PCR product spanning coding regions 6 and 7 of TYRP1. Presence of this PCR product is sufficient to conclude that these animals have an intact copy of coding regions 6 and 7 on at least one chromosome. We have added an explanation of the assay, as follows,

“These primers reside within the putative deletion and therefore cannot distinguish between animals lacking the putative deletion and animals heterozygous for it. Presence of a PCR product in this assay indicates that TYRP1 coding regions 6 and 7 are present on at least one chromosome. Absence of a PCR product in this assay indicates that TYRP1 coding regions 6 and 7 are absent on both chromosomes. We were unable to design a PCR assay to detect the putative deletion in heterozygotes because the putative deletion extends into a discontinuity in the Burmese python reference genome. Presence or absence of a PCR product in this assay is provided in S6 Table.”

L337-339 remove the parentheses

Thank you for the suggestion. We have revised the text accordingly.

L386 "slicing" should read splicing

Thank you for finding this mistake. We have fixed it.

L447 Current state-of-the-art methods (such as targeted high-throughput sequencing, whole-genome sequencing, or long-read technologies) should be mentioned here as an obvious solution to missing reference and/or structural variants. See (cite) e.g. https://www.mdpi.com/2073-4425/10/5/386/htm for an example of a coat color gene variant identification without correct reference sequence.

We agree that long-read technologies provide the most straightforward avenue for improving genome assemblies in pythons and identifying structural variants. We have revised the text to include this point. The text reads,

“As genomic resources for Burmese python and ball python expand, ideally with the use long-read technologies to improve genome assemblies and identify structural variants, we expect to identify a putatively causative variant on the TYRAlbino haplotype and to map the breakpoints of the putative deletion in TYRP1.”

L454 "strip" should be stripes

Thank you for finding this mistake. We have fixed it.

L485 "supports" at the end of sentence should probably be deleted

Thank you for finding this mistake. We have fixed it.

L487 How many were "Toffee" or "Candy"?

We have revised the text accordingly. It now reads,

“The Albinos were composed of 36 animals described as AlbAlbino homozygotes, three animals described as AlbCandy homozygotes, one animal described as an AlbToffee homozygote, nine animals described as AlbAlbino / AlbCandy compound homozygotes, and one animal described as an AlbAlbino / AlbToffee compound homozygote.”

L525 Please, specify the software used. Does TBLASTN refer to the NIH Translated BLAST tool? Citation should be provided, as well as clarification, if default parameters were used for the search or if anything was adjusted.

We have added a brief explanation of TBLASTN, as well as citations for it. We have also specified that TBLASTN searches were performed using default parameters. This text reads,

“To curate new gene annotations, we performed TBLASTN [110,111] searches of the Burmese python genome (Python_molurus_bivittatus-5.0.2) using protein sequences from anole lizard or corn snake as the query. TBLASTN is a search tool that aligns a query protein sequence all six reading frames of a nucleotide database. TBLASTN searches were performed using default parameters.”

---

## [Decision Letter · Decision Letter 3]

6 Oct 2022

A community-science approach identifies genetic variants associated with three color morphs in ball pythons (*Python regius*)

PONE-D-21-16582R3

Dear Dr. Seidel,

We’re pleased to inform you that your manuscript has been judged scientifically suitable for publication and will be formally accepted for publication once it meets all outstanding technical requirements.

Kind regards,

Xiaolin Bi, Ph.D.

Academic Editor

PLOS ONE

Additional Editor Comments (optional):

Reviewers' comments:

Reviewer's Responses to Questions

**Comments to the Author**

1. If the authors have adequately addressed your comments raised in a previous round of review and you feel that this manuscript is now acceptable for publication, you may indicate that here to bypass the “Comments to the Author” section, enter your conflict of interest statement in the “Confidential to Editor” section, and submit your "Accept" recommendation.

Reviewer #3: All comments have been addressed

2. Is the manuscript technically sound, and do the data support the conclusions?

Reviewer #3: Yes

3. Has the statistical analysis been performed appropriately and rigorously? 

Reviewer #3: N/A

4. Have the authors made all data underlying the findings in their manuscript fully available?

Reviewer #3: Yes

5. Is the manuscript presented in an intelligible fashion and written in standard English?

Reviewer #3: Yes

6. Review Comments to the Author

Reviewer #3: (No Response)

7. PLOS authors have the option to publish the peer review history of their article (what does this mean?). If published, this will include your full peer review and any attached files.

Reviewer #3: No

---

## [Editor Report · Acceptance letter]

10 Oct 2022

PONE-D-21-16582R3 

A community-science approach identifies genetic variants associated with three color morphs in ball pythons (*Python regius*) 

Dear Dr. Seidel:

I'm pleased to inform you that your manuscript has been deemed suitable for publication in PLOS ONE. Congratulations! Your manuscript is now with our production department. 

Kind regards, 

on behalf of

Dr. Xiaolin Bi 

Academic Editor

PLOS ONE